# Towards Understanding Valuable Preference Data for Large Language Model Alignment

**Zizhuo Zhang**[1][*]  **Qizhou Wang**[1]  **Shanshan Ye**[2][†]  **Jianing Zhu**[1]  **Jiangchao Yao**[3]
**Bo Han**[1,5][†]  **Masashi Sugiyama**[4,5]
[1]TMLR Group, Department of Computer Science, Hong Kong Baptist University
[2]University of Technology Sydney    [3]CMIC, Shanghai Jiao Tong University
[4]The University of Tokyo    [5]RIKEN Center for Advanced Intelligence Project
{cszzzhang, csqzwang, csjnzhu, bhanml}@comp.hkbu.edu.hk
shanshan.ye@student.uts.edu.au, Sunarker@sjtu.edu.cn
sugi@k.u-tokyo.ac.jp

## Abstract

Large language model (LLM) alignment is typically achieved through learning from human preference comparisons, making the quality of preference data critical to its success. Existing studies often pre-process raw training datasets to identify valuable preference pairs using external reward models or off-the-shelf LLMs, achieving improved overall performance but rarely examining whether individual, selected data point is genuinely beneficial. We assess data quality through individual influence on validation data using our newly proposed truncated influence function (TIF), which mitigates the over-scoring present in traditional measures and reveals that preference data quality is inherently a property of the model. In other words, a data pair that benefits one model may harm another. This leaves the need to improve the preference data selection approaches to be adapting to specific models. To this end, we introduce two candidate scoring functions (SFs) that are computationally simpler than TIF and positively correlated with it. They are also model dependent and can serve as potential indicators of individual data quality for preference data selection. Furthermore, we observe that these SFs inherently exhibit errors when compared to TIF. To this end, we combine them to offset their diverse error sources, resulting in a simple yet effective data selection rule that enables the models to achieve a more precise selection of valuable preference data. We conduct experiments across diverse alignment benchmarks and various LLM families, with results demonstrating that better alignment performance can be achieved using less data, showing the generality of our findings and new methods. Our code is publicly available at https://github.com/tmlr-group/TIF_LossDiff-IRM.

## 1 Introduction

Reinforcement learning with human feedback (RLHF) has emerged as a dominant fine-tuning paradigm for aligning large language models (LLMs) with human preferences (Bai et al., 2022b; Wang et al., 2025b). Whether through training explicit reward models (Lambert et al., 2024) or optimizing policy with implicit rewards (Rafailov et al., 2023), the success of RLHF hinges heavily on the availability of high-quality preference data. Previous works (Shen et al., 2024; Pattnaik et al., 2024; Morimura et al., 2024; Deng et al., 2025; Chen et al., 2024) typically leverage external reward models or off-the-shelf LLMs to filter raw data, treating selected data as reliable sources for training their models or releasing them for open-source use (Cui et al., 2023; Bai et al., 2022a). This pre-processing paradigm dominates the RLHF community to improve data quality, contributing remarkably to the success of many publicly available LLMs such as Llama (Vavekanand & Sam, 2024), Qwen (Yang et al., 2025) and DeepSeek (Guo et al., 2025).

---

[*]Trainee at RIKEN AIP.
[†]Correspondence to Shanshan Ye (shanshan.ye@student.uts.edu.au) and Bo Han (bhanml@comp.hkbu.edu.hk).

However, such pre-processing implicitly assumes that quality is an intrinsic property of data themselves: regardless of training configurations or models, certain data are consistently presumed to be more valuable for alignment than others. Therefore, a seemingly more realistic perspective is that the data quality is also a property of the model (Grosse et al., 2023; Xia et al., 2024). That is to say, some data points may be beneficial for certain models or configurations while being detrimental at others. We validate this new assumption as more reasonable based on the influence function (IF) (Koh & Liang, 2017), which quantifies the impact of each training data point on validation performance, thereby reflecting data quality. Nevertheless, we observe that the influence scores may overfit to the validation data, an issue that is particularly severe for open-world models like LLMs. To address this, we propose and verify a simple modification to the original IF, namely *truncated influence function* (TIF), which is shown to be more reliable for preference data selection and results in better overall performance. Using TIF, we verify that data quality is model-dependent, varying across different models, with some data points proving beneficial for certain models yet harmful at others.

Our above analysis suggests a reasonable yet seldom-discussed viewpoint: preference data selection should be performed for specific models and explicitly related to the training process. Although TIF provides a reliable and effective measure of data quality, its high computational cost on gradients limits its direct application for large-scale LLMs (Kwon et al., 2024). To address this, we introduce two simpler scoring functions (SFs) with lower computational costs yet sufficient potential to approximate TIF–*loss difference* (LossDiff) and *implicit reward margin* (IRM)–that are positively correlated with TIF yet require only forward passes. Considering that a single SF may exhibit specific errors relative to TIF, a combination indicator *LossDiff-IRM* is proposed to mitigates such errors, as their distinct bias sources may offset one another. Empirically, LossDiff-IRM achieves an average WinRate improvement of $+13.58\%$ over full-data training while using only 50%–64% of the data, across multiple LLM families, benchmarks, and alignment methods.

## 2 PRELIMINARY

To begin, we formalize pairwise preference data and introduce Direct Preference Optimization (DPO) as the base preference optimization approach used in our analysis and experiments, and then briefly review prior work on preference data selection.

**Pairwise Preference Data.** Pairwise preference dataset, denoted as $\mathcal{D} = \{d_i = (x^{(i)}, y_w^{(i)}, y_l^{(i)})\}_{i=1}^N$, is annotated by humans to reflect real human preference. Each pair consists of a prompt $x$ and a pair of responses: the chosen response $y_w$ and the rejected response $y_l$, which means that the human annotator prefers the chosen response rather than the rejected one, denoted as $y_w \succ y_l$.

**Direct Preference Alignment.** Traditional RLHF (Bai et al., 2022b) often follows a two-stage pipeline: first training a reward model on preference data, and then optimizing the policy LLM using reinforcement learning (RL) algorithm such as PPO (Schulman et al., 2017). The two-stage RLHF is relatively complicated and resource-intensive. Recently, many studies (Rafailov et al., 2023; Azar et al., 2024; Zhao et al., 2023; Meng et al., 2024; Wu et al., 2024b;a) have tried to bypass the need for an explicit reward model and RL learning, to directly optimize the policy model from preference data. Among them, DPO (Rafailov et al., 2023) is a milestone work that derives the optimal policy and substitutes it into the Bradley–Terry (BT) model (Bradley & Terry, 1952), which yields its objective:

$$\mathcal{L}_{\text{DPO}}(\theta; \mathcal{D}) = -\mathbb{E}_{(x,y_w,y_l)\in\mathcal{D}} \left[ \log \sigma \left( \beta \log \frac{\pi_\theta(y_w|x)}{\pi_{\text{ref}}(y_w|x)} - \beta \log \frac{\pi_\theta(y_l|x)}{\pi_{\text{ref}}(y_l|x)} \right) \right], \quad (1)$$

where $\beta > 0$ controls the strength of the KL penalty between $\pi_\theta$ and $\pi_{\text{ref}}$, and $\sigma$ is the sigmoid function. According to the derivation of DPO loss (Rafailov et al., 2023), the term inside the sigmoid can be interpreted as an *implicit reward margin* (IRM) between the chosen and rejected responses:

$$\text{IRM}_\theta(d) = \beta \log \frac{\pi_\theta(y_w|x)}{\pi_{\text{ref}}(y_w|x)} - \beta \log \frac{\pi_\theta(y_l|x)}{\pi_{\text{ref}}(y_l|x)}. \quad (2)$$

This reward margin serves as an important indicator monitored during training to track how well the model differentiates the chosen response from the rejected one. For an individual data pair, the margin is negatively correlated with the DPO loss: larger margins correspond to lower loss values. Intuitively, the DPO adopts a contrastive-like structure that directly trains the policy LLM on preference pairs by

encouraging a larger relative reward margin for the chosen response over the rejected one. The other alignment method SLiC (Zhao et al., 2023) used in this work is introduced in Appendix B.1.

**Preference Data Quality.** Given annotator subjectivity and the open nature of the concept of preference, manually annotated preference data can be imperfect or noisy. Existing studies adopt multiple data processing strategies: Morimura et al. (2024) and Deng et al. (2025) filter out low-quality preference pairs based on external reward models. Pattnaik et al. (2024) organizes the preference data in a curriculum based on metrics such as GPT-4 score, external reward score, or log probability. Muldrew et al. (2024) and Shen et al. (2025) introduce active learning to improve data quality and annotation efficiency. Overall, these approaches mainly rely on external signals (e.g., GPT or reward model scores, or active human labeling) and implicitly view preference data quality as a property of the data itself, while overlooking the role of models, training configurations, and optimization objectives in shaping utility of preference data. Detailed related work refers to Appendix A.2.

## 3 ANALYSIS: TRUNCATED INFLUENCE FUNCTION (TIF)

Building upon influence function (IF), this section takes a model-centric perspective to investigate what kind of preference data truly valuable for model alignment.

### 3.1 INFLUENCE FUNCTION

To quantify the quality of a data sample $d \in \mathcal{D}_{\text{train}}$, a classical idea is to measure its leave-one-out (LOO) (Evgeniou et al., 2004; Elisseeff et al., 2003) effect, which assesses the change in validation performance when the model is trained with versus without the data sample $d$:

$$\text{LOO Effect}(d) = v\left(\theta_{\mathcal{D}_{\text{train}}}; \mathcal{D}_{\text{val}}\right) - v\left(\theta_{\mathcal{D}_{\text{train}} \setminus \{d\}}; \mathcal{D}_{\text{val}}\right), \tag{3}$$

where $v(\cdot)$ denotes a measurement to evaluate the model $\pi_\theta$, and $\theta_{\mathcal{D}}$ denotes the model trained on the dataset $\mathcal{D}$. Intuitively, computing the exact LOO effect requires training $|\mathcal{D}_{\text{train}}| + 1$ separate models, which is infeasible in practice. To avoid repetitive retraining the model, *influence function* (IF) (Koh & Liang, 2017) provides a first-order Taylor approximation of the LOO effect using a gradient-based metric at the current parameters:

$$\text{IF}(d; \pi_\theta; \mathcal{D}_{\text{val}}) = \nabla_\theta \mathcal{L}(\theta; \mathcal{D}_{\text{val}})^\top H_\theta^{-1} \nabla_\theta \ell(\theta; d) \approx \nabla_\theta \mathcal{L}(\theta; \mathcal{D}_{\text{val}})^\top \nabla_\theta \ell(\theta; d) \tag{4}$$

$$= \left(\frac{1}{|\mathcal{D}_{\text{val}}|} \sum_{i=1}^{|\mathcal{D}_{\text{val}}|} \nabla_\theta \ell\left(\theta; d_{\text{val}}^{(i)}\right)\right)^\top \nabla_\theta \ell(\theta; d), \tag{5}$$

where $\ell(\theta; d)$ is the training loss for the data $d$, and $H_\theta := \nabla_\theta^2 \mathcal{L}(\theta; \mathcal{D}_{\text{train}})$ is the Hessian matrix (Hampel, 1974) of the total training loss with respect to model parameters $\theta$. In practice, computing and inverting the Hessian matrix is often computationally intractable, especially for a large-scale model. A common approach is to approximate the Hessian $H_\theta$ as the identity matrix by assuming that the loss landscape is locally isotropic near $\theta$ (Koh & Liang, 2017; Wang et al., 2025a; Xia et al., 2024). This simplification reduces the IF to a dot product between the gradient of the training data $d$ and the expected gradient of the validation set. Intuitively, a higher IF means the training gradient better aligns with the validation gradient, suggesting the data is more beneficial for generalization.

**Instantiation to DPO Loss.** By taking the DPO loss in Eq. (1) into the IF formulation in Eq. (4), we instantiate the IF under DPO objective for a give preference pair $d = (x, y_w, y_l)$:

$$\text{IF}_{\text{DPO}}(d; \pi_\theta; \mathcal{D}_{\text{val}}) := \beta(1 - \sigma(\Delta_\theta)) \left\langle \underbrace{\frac{\beta}{|\mathcal{D}_{\text{val}}|} \sum_i \left(1 - \sigma(\Delta_\theta^{(i)})\right) \left(g_w^{(i)} - g_l^{(i)}\right)}_{\substack{\text{preference generalization direction} \\ \text{w.r.t. validation set}}}, \underbrace{g_w - g_l}_{\substack{\text{current preference} \\ \text{pair direction}}} \right\rangle, \tag{6}$$

where $g_* = \nabla_\theta \log \pi_\theta(y_*|x)$ denotes the gradient of the log-likelihood for response $y_* \in \{y_w, y_l\}$, and $\Delta_\theta = \beta \log \frac{\pi_\theta(y_w|x)}{\pi_{\text{ref}}(y_w|x)} - \beta \log \frac{\pi_\theta(y_l|x)}{\pi_{\text{ref}}(y_l|x)}$ is the reward difference term in the DPO loss. Unlike IF under pointwise objectives, which assesses the gradient consistency of a give data point with validation set (Koh & Liang, 2017; Wang et al., 2025a), the DPO-based IF focuses on the gradient

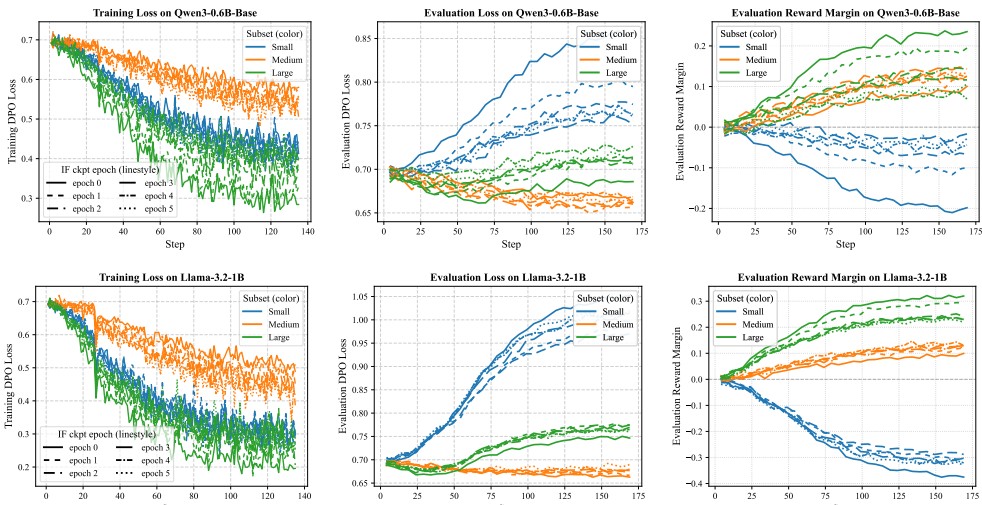

Figure 1: **Analysis of IF-based data partition on the Qwen3-0.6B-Base and Llama-3.2-1B.** *From Left to Right:* Training DPO loss, evaluation DPO loss and evaluation reward margin. Subsets of {small, medium, large} are denoted by {blue, orange, green}, respectively, while different line styles indicate IF values computed using different epoch checkpoints. More analysis refers to Appendix C.1.

difference consistency, i.e., $g_w - g_l$, with the validation preferences, thereby serving as a proxy to assess quality of a preference pair $d$. Vanilla IF implicitly assumes the validation set is an oracle of generalization, thereby equating high-IF data with high quality. However, under this assumption, IF may overfit to the specific validation set. The instantiation to SLiC loss is provided in Appendix B.1.

## 3.2 ANALYSIS: TRUNCATED INFLUENCE FUNCTION (TIF)

Since preference alignment is open-ended and a fully reliable validation set is hard to obtain, we propose the Truncated Influence Function (TIF) by partitioning data into three regions for analysis.

**Analytical Setup.** We randomly sample a probing set $\mathcal{D}^{\mathrm{prob}} = \{\mathcal{D}^{\mathrm{prob}}_{\mathrm{train}}, \mathcal{D}^{\mathrm{prob}}_{\mathrm{val}}, \mathcal{D}^{\mathrm{prob}}_{\mathrm{test}}\}$ from UltraFeed-back (Cui et al., 2023), containing 5,000, 3,000, 300 preference pairs for training, validation, and testing, respectively. Qwen3-0.6B-Base (Yang et al., 2025) and Llama-3.2-1B (Vavekanand & Sam, 2024) are adopted as backbone models. Both models are first trained with supervised fine-tuning (SFT) on UltraChat-200K (Ding et al., 2023), followed by five epochs of DPO on $\mathcal{D}^{\mathrm{prob}}_{\mathrm{train}}$. At each-epoch DPO checkpoint, we compute the IF values of all training pairs using Eq. (5) with respect to $\mathcal{D}^{\mathrm{prob}}_{\mathrm{val}}$ and partition them into three equally sized subsets (small-, medium-, and large-IF). Each subset is then used to continue DPO training from the checkpoint, while we monitor the evaluation DPO loss and reward margin on $\mathcal{D}^{\mathrm{prob}}_{\mathrm{test}}$ to examine which type of data is more beneficial for alignment training.

Figure 1 reports the training dynamics on Qwen3-0.6B-Base and Llama-3.2-1B with different IF-based partitions. Across both models, we observe several similar phenomena as follows:

- **Small-IF data:** Training loss decreases as expected, but evaluation loss increases while the evaluation reward margin falls below zero. This indicates that small-IF pairs are largely uninformative and of low quality, with a high likelihood of being noisy or ambiguous. Learning from such data not only fails to help the model distinguish chosen from rejected responses but may also mislead it, thereby harming alignment training and preference generalization.

- **Large-IF data:** Evaluation loss initially decreases but later rises, while the evaluation reward margin continues to increase. This mismatch between loss and margin trajectories is counterintuitive, since DPO loss and reward margin are normally negatively correlated at the per-sample level. The phenomenon indicates overfitting: training on large-IF data enlarges the margins of a small subset of pairs while reducing the margins of many other pairs. Owing to the sigmoid saturation, overly large margins contribute little to the loss, whereas the diminished margins of the majority dominate it, ultimately causing the model to overfit to a narrow portion of preference pairs.

Table 1: **Computational time and throughput rate analysis** on probing set by Qwen3-0.6B-Base and Llama-3.2-1B using one H100-80GB GPU. *Top:* Exact IF, consisting of the validation-gradient computation and the per-pair IF inner-product computation. *Bottom:* LossDiff-IRM, consisting of one forward on the training model and one forward on a validation-aligned auxiliary model.

| | Computational Time | | | Throughput Rate (pair/sec) | |
|---|---|---|---|---|---|
| **IF Computation** | Val Gradient | IF | Total | Val Gradient | IF |
| Qwen3-0.6B-Base | 1 h 17 m 31 s | 1 h 59 m 45 s | 3 h 17 m 16 s | 1.55 | 1.44 |
| Llama-3.2-1B | 3 h 55 m 32 s | 6 h 02 m 37 s | 9 h 58 m 09 s | 4.71 | 4.35 |
| **LossDiff-IRM Computation** | Training Forward | Val Forward | Total | Training Forward | Val Forward |
| Qwen3-0.6B-Base | 1 min 5 s | 59 s | 2 m 4 s | 76.92 | 84.74 |
| Llama-3.2-1B | 2 m 32 s | 2 m 27 s | 4 m 59 s | 32.86 | 33.80 |

Table 2: **Overlap Coefficient analysis** to assess the overlap of two selected sets.

| Overlap Coefficient | LossDiff vs. IF | | IRM vs. IF | | LossDiff-IRM vs. IF | |
|---|---|---|---|---|---|---|
| **Models** | Epoch 1 ckpt | Epoch 2 ckpt | Epoch 1 ckpt | Epoch 2 ckpt | Epoch 1 ckpt | Epoch 2 ckpt |
| Qwen3-0.6B-Base | 0.6953 | 0.6639 | 0.6883 | 0.6470 | **0.7820** | **0.7257** |
| Llama-3.2-1B | 0.6687 | 0.6582 | 0.6969 | 0.6025 | **0.7657** | **0.6963** |

- **Medium-IF data:** As training loss decreases, evaluation loss steadily decreases while reward margin increases, which aligns well with the intended learning dynamics of the DPO objective: improving the model ability to distinguish human-preferred (chosen) responses from rejected ones. This demonstrates that medium-IF preference pairs are high-quality data, providing the most effective and stable signal for alignment training and fostering better preference generalization.

Overall, these findings differ from vanilla IF in traditional classification, where high-IF data is typically regarded as the most valuable. In preference alignment, however, the most valuable data are the medium-IF preference pairs. This counter-intuitive finding is nonetheless reasonable: preference alignment is an open-ended task in which the chosen–rejected annotations inherently affected by annotator subjectivity, making the validation gradient an imperfect proxy for reflecting the real human preference direction. As a result, data with both extremely small and extremely large IF values is low-quality, which provide little useful signal for alignment training. Therefore, we propose the Truncated Influence Function (TIF), which offers a more robust criterion for assessing preference data quality in LLM alignment training:

$$\text{TIF}(d; \pi_\theta; \mathcal{D}_{\text{val}}) = \mathbb{I}\left[\delta_{\text{small}} < \text{IF}(d; \pi_\theta; \mathcal{D}_{\text{val}}) < \delta_{\text{large}}\right], \tag{7}$$

where $\delta_{\text{small}}$ and $\delta_{\text{large}}$ denote threshold percentiles that specify the boundaries of IF values. It can be observed that the criterion of TIF depends on the current model $\pi_\theta$, which suggests that the identification of valuable preference pairs is inherently model-dependent.

## 4 METHODOLOGY: LOSSDIFF-IRM DATA SELECTION

Although TIF provides a principled criterion for assessing the value of preference data, its computation requires gradients on both the training and validation sets, which becomes prohibitive in the large-scale model and dataset regime. As reported in the top of Table 1, computing exact IF on a probing set of 5,000 pairs for a small Llama-3.2-1B model still takes about 10 hours, which is already prohibitively slow. This computational cost makes TIF impractical as a per-sample scoring function for preference data selection in alignment training at scale.

### 4.1 APPROXIMATION PROXY OF TIF

To address this challenge, we introduce lightweight, *model-dependent* indicators that require only forward passes yet track TIF well. Concretely, we use two approximation proxies that are positively correlated with IF at the per-pair level for efficient preference data selection as follows:

**Validation-based scoring function: Loss Difference (LossDiff).** To bypass costly gradient computations, we introduce an auxiliary model that is aligned on the validation set and approximate IF by the *loss difference* between the current model and this auxiliary model. Specifically, for each preference pair, we define its loss difference as follows:

$$\text{LossDiff}(d; \pi_\theta, \pi_{\theta_{\text{val}}}) = \ell(\theta; d) - \ell(\theta_{\text{val}}; d), \tag{8}$$

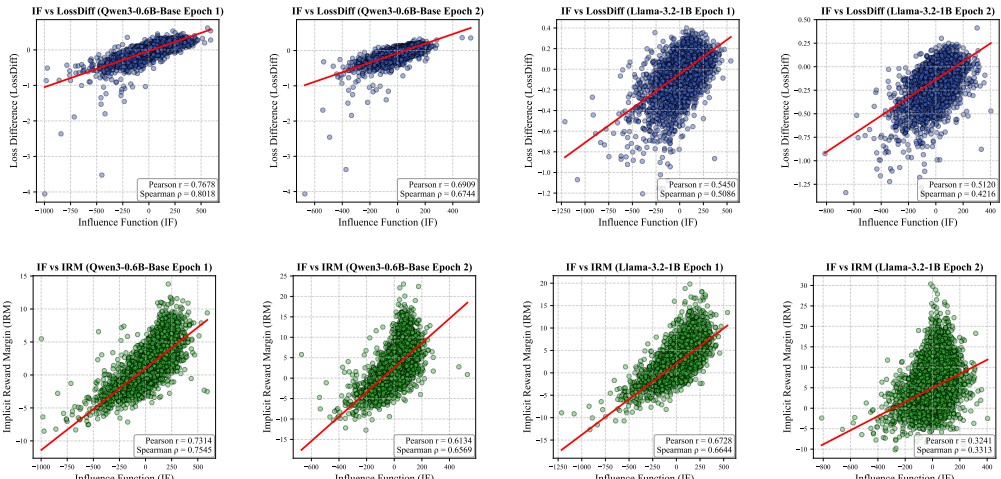

Figure 2: **Correlation analysis on Qwen3-0.6B-Base and Llama-3.2-1B.** *Top:* correlation between loss difference (LossDiff) and IF. *Bottom:* correlation between implicit reward margin (IRM) and IF.

where $\pi_{\theta_{\text{val}}}$ is the auxiliary model aligned on validation set. An intuitive understanding is that a larger value indicates that moving from $\theta$ toward $\theta_{\text{val}}$ reduces the loss on $d$, which is consistent with the direction favored by the validation objective. We formally demonstrate that the LossDiff has a positive correlation with the IF in Appendix B.2. Empirically, the top panel of Figure 2 shows strong Pearson (Cohen et al., 2009) and Spearman (Hauke & Kossowski, 2011) correlations (e.g., $r = 0.77$, $\rho = 0.80$ on Qwen-0.6B-Base). Notably, this proxy requires only two forward passes per pair without backpropagation and remains tied to the validation set by construction.

**Validation-free scoring function: Implicit Reward Margin (IRM).** Empirically, we find that the IRM defined in Eq. (2) exhibits a strong positive correlation with IF at the per-pair level. As illustrated in the bottom panel of Figure 2, the Pearson and Spearman correlations between IRM and IF reach $r = 0.67$ and $\rho = 0.66$ on Llama-3.1-1B, respectively. Intuitively, IRM measures how strongly the current model $\pi_\theta$ prefers the chosen response over the rejected response relative to a reference, and thus reflects the *model-perceived difficulty* of a preference pair. When training on a pair, the update mainly pushes the policy to enlarge that pair's margin; the validation objective is doing the same at the validation distribution level. Consequently, pairs with larger positive IRM tend to produce updates that are more consistent with the validation objective and thus yield larger IF. Moreover, IRM is a validation-free scoring function and requires only forward computation from $\pi_\theta$.

From the Pearson and Spearman correlations, we also observe that the validation-based LossDiff correlates higher with IF than the validation-free IRM. This is expected, as LossDiff explicitly leverages a validation-aligned auxiliary model, with the validation set serving as a reference direction for human preference. In contrast, IRM relies solely on the internal signals of current model, which makes it more lightweight but also less anchored. More analysis is provided in Appendix C.2.

## 4.2 LossDiff-IRM Preference Data Selection

Using scoring function of either LossDiff or IRM alone may introduce method-specific errors. To offset the diverse errors, we propose a combined indicator: *LossDiff–IRM*, which selects data falling within the intersection of the medium percentile ranges defined by LossDiff and IRM. Specifically, LossDiff-IRM selects a preference pair $d$ if and only if

$$\begin{aligned}
\text{LossDiff–IRM}(d; \pi_\theta; \mathcal{D}_{\text{val}}) = {} &\mathbb{I}[\xi_{\text{small}} < \text{LossDiff}(d; \pi_\theta; \mathcal{D}_{\text{val}}) < \xi_{\text{large}}] \\
&\wedge \ \mathbb{I}[\tau_{\text{small}} < \text{IRM}(d; \pi_\theta) < \tau_{\text{large}}].
\end{aligned} \tag{9}$$

where $\text{LossDiff-IRM}(d; \pi_\theta; \mathcal{D}_{\text{val}}) \in \{0, 1\}$, $(\xi_{\text{small}}, \xi_{\text{large}})$ and $(\tau_{\text{small}}, \tau_{\text{large}})$ are percentile thresholds that define the medium ranges for LossDiff and IRM, respectively. The procedure is as follows: we first warm up $\pi_\theta$ by training for a short stage (e.g., one epoch) on the training set, then obtain $\pi_{\theta_{\text{val}}}$ by training for one stage on the validation set. We then compute LossDiff using both $\pi_\theta$ and $\pi_{\theta_{\text{val}}}$, and IRM using $\pi_\theta$. Based on the combined rule in Eq. (9), we select preference data and retrain $\pi_\theta$ for a longer stage (e.g., two epochs in our experiments).

Table 3: **Performance of LossDiff-IRM and baselines using DPO and SLiC.** Cell background colors indicate relative performance: darker colors denote better results within each model group.

**Llama-3.1-8B (DPO)**

| Methods | Dataset Ratio | UltraFeedback Single ↑ | WinRate ↑ | AlpacaEval Single ↑ | WinRate ↑ | Vicuna-Bench Single ↑ | WinRate ↑ | Arena-Hard Single ↑ | WinRate ↑ |
|---|---|---|---|---|---|---|---|---|---|
| SFT | | 3.60 | - | 3.53 | - | 3.98 | - | 2.63 | - |
| Full Data | 100% | 5.77 | 77.61 | 5.87 | 78.41 | 6.04 | 73.75 | 4.68 | 81.39 |
| Random | 64% | 5.52 | 74.83 | 5.59 | 75.93 | 5.46 | 68.13 | 4.64 | 81.27 |
| GPT4 | 64% | 6.04 | 80.57 | 6.21 | 81.09 | 6.86 | 80.31 | 4.96 | 84.30 |
| Reward Model | 64% | 6.24 | 82.68 | 6.38 | 83.76 | 6.45 | 81.76 | 5.13 | 86.19 |
| LossDiff-IRM | 64% | 6.54 | 83.97 | 6.84 | 87.08 | 7.06 | 86.88 | 5.59 | 88.40 |

**Qwen3-8B-Base (DPO)**

| Methods | Dataset Ratio | UltraFeedback Single ↑ | WinRate ↑ | AlpacaEval Single ↑ | WinRate ↑ | Vicuna-Bench Single ↑ | WinRate ↑ | Arena-Hard Single ↑ | WinRate ↑ |
|---|---|---|---|---|---|---|---|---|---|
| SFT | | 6.97 | - | 6.88 | - | 7.94 | - | 6.64 | - |
| Full Data | 100% | 7.64 | 61.41 | 7.92 | 63.85 | 8.21 | 62.14 | 7.58 | 59.61 |
| Random | 64% | 7.71 | 61.47 | 7.94 | 64.12 | 8.26 | 58.93 | 7.57 | 62.07 |
| GPT4 | 64% | 7.69 | 62.19 | 8.01 | 63.81 | 8.28 | 52.19 | 7.62 | 61.53 |
| Reward Model | 64% | 7.81 | 64.19 | 8.24 | 69.35 | 8.56 | 66.25 | 7.61 | 64.78 |
| LossDiff-IRM | 64% | 8.05 | 67.32 | 8.36 | 71.52 | 8.72 | 67.19 | 7.83 | 68.63 |

**Pythia-2.8B (DPO)**

| Methods | Dataset Ratio | UltraFeedback Single ↑ | WinRate ↑ | AlpacaEval Single ↑ | WinRate ↑ | Vicuna-Bench Single ↑ | WinRate ↑ | Arena-Hard Single ↑ | WinRate ↑ |
|---|---|---|---|---|---|---|---|---|---|
| SFT | | 3.94 | - | 4.35 | - | 4.66 | - | 2.74 | - |
| Full Data | 100% | 4.60 | 70.53 | 4.95 | 67.05 | 5.35 | 67.50 | 2.97 | 60.71 |
| Random | 64% | 4.54 | 68.27 | 4.79 | 64.03 | 5.15 | 68.13 | 3.00 | 63.25 |
| GPT4 | 64% | 4.71 | 72.02 | 4.96 | 67.10 | 5.39 | 73.75 | 3.08 | 64.15 |
| Reward Model | 64% | 4.68 | 75.73 | 5.09 | 70.91 | 5.60 | 75.95 | 3.03 | 63.03 |
| LossDiff-IRM | 64% | 4.90 | 79.62 | 5.30 | 76.03 | 5.74 | 82.50 | 3.26 | 71.64 |

**Pythia-1.4B (DPO)**

| Methods | Dataset Ratio | UltraFeedback Single ↑ | WinRate ↑ | AlpacaEval Single ↑ | WinRate ↑ | Vicuna-Bench Single ↑ | WinRate ↑ | Arena-Hard Single ↑ | WinRate ↑ |
|---|---|---|---|---|---|---|---|---|---|
| SFT | | 3.50 | - | 3.65 | - | 4.20 | - | 2.37 | - |
| Full Data | 100% | 3.70 | 65.88 | 3.99 | 66.17 | 4.71 | 64.38 | 2.65 | 60.24 |
| Random | 52% | 3.78 | 67.43 | 4.05 | 68.16 | 4.56 | 61.25 | 2.60 | 61.64 |
| GPT4 | 52% | 3.96 | 70.75 | 4.28 | 70.96 | 4.89 | 68.75 | 2.83 | 64.20 |
| Reward Model | 52% | 3.83 | 70.80 | 4.18 | 70.83 | 4.84 | 68.75 | 2.71 | 64.76 |
| LossDiff-IRM | 52% | 4.23 | 78.49 | 4.49 | 76.43 | 5.28 | 76.88 | 2.96 | 71.72 |

**Pythia-410M (DPO)**

| Methods | Dataset Ratio | UltraFeedback Single ↑ | WinRate ↑ | AlpacaEval Single ↑ | WinRate ↑ | Vicuna-Bench Single ↑ | WinRate ↑ | Arena-Hard Single ↑ | WinRate ↑ |
|---|---|---|---|---|---|---|---|---|---|
| SFT | | 2.56 | - | 2.47 | - | 3.15 | - | 1.91 | - |
| Full Data | 100% | 2.81 | 75.25 | 2.77 | 73.51 | 3.10 | 69.37 | 2.06 | 59.47 |
| Random | 56% | 2.94 | 76.03 | 2.92 | 76.62 | 3.58 | 70.63 | 2.06 | 57.08 |
| GPT4 | 56% | 2.95 | 76.78 | 2.95 | 79.81 | 3.49 | 79.37 | 2.15 | 61.44 |
| Reward Model | 56% | 2.96 | 81.48 | 3.02 | 80.64 | 3.67 | 74.38 | 2.16 | 60.59 |
| LossDiff-IRM | 56% | 3.30 | 86.14 | 3.30 | 85.16 | 3.80 | 85.62 | 2.38 | 69.63 |

**Llama-3.1-8B (SLiC)**

| Methods | Dataset Ratio | UltraFeedback Single ↑ | WinRate ↑ | AlpacaEval Single ↑ | WinRate ↑ | Vicuna-Bench Single ↑ | WinRate ↑ | Arena-Hard Single ↑ | WinRate ↑ |
|---|---|---|---|---|---|---|---|---|---|
| SFT | | 3.60 | - | 3.53 | - | 3.98 | - | 2.63 | - |
| Full Data | 100% | 5.09 | 70.72 | 5.13 | 72.13 | 5.40 | 71.88 | 3.98 | 73.75 |
| Random | 64% | 4.94 | 69.52 | 4.89 | 67.05 | 5.26 | 67.50 | 3.95 | 70.35 |
| GPT4 | 64% | 5.48 | 75.61 | 5.40 | 72.89 | 6.05 | 67.81 | 4.28 | 75.27 |
| Reward Model | 64% | 5.34 | 73.64 | 5.54 | 75.03 | 5.55 | 68.75 | 4.50 | 78.32 |
| LossDiff-IRM | 64% | 5.94 | 79.51 | 5.84 | 78.84 | 5.85 | 76.56 | 4.65 | 83.12 |

**Qwen3-8B-Base (SLiC)**

| Methods | Dataset Ratio | UltraFeedback Single ↑ | WinRate ↑ | AlpacaEval Single ↑ | WinRate ↑ | Vicuna-Bench Single ↑ | WinRate ↑ | Arena-Hard Single ↑ | WinRate ↑ |
|---|---|---|---|---|---|---|---|---|---|
| SFT | | 6.97 | - | 6.88 | - | 7.94 | - | 6.64 | - |
| Full Data | 100% | 7.55 | 59.54 | 7.61 | 59.71 | 8.05 | 54.69 | 7.21 | 59.61 |
| Random | 64% | 7.57 | 57.76 | 7.63 | 62.05 | 7.97 | 47.94 | 7.25 | 59.18 |
| GPT4 | 64% | 7.64 | 59.91 | 7.89 | 63.62 | 8.21 | 58.37 | 7.33 | 59.47 |
| Reward Model | 64% | 7.74 | 62.47 | 8.09 | 66.57 | 8.50 | 63.44 | 7.38 | 62.27 |
| LossDiff-IRM | 64% | 7.87 | 64.40 | 8.11 | 67.58 | 8.44 | 61.12 | 7.61 | 62.20 |

**Pythia-2.8B (SLiC)**

| Methods | Dataset Ratio | UltraFeedback Single ↑ | WinRate ↑ | AlpacaEval Single ↑ | WinRate ↑ | Vicuna-Bench Single ↑ | WinRate ↑ | Arena-Hard Single ↑ | WinRate ↑ |
|---|---|---|---|---|---|---|---|---|---|
| SFT | | 3.94 | - | 4.35 | - | 4.66 | - | 2.74 | - |
| Full Data | 100% | 4.36 | 67.46 | 4.48 | 61.66 | 4.90 | 65.00 | 2.93 | 59.04 |
| Random | 64% | 4.31 | 63.00 | 4.56 | 59.73 | 5.16 | 65.62 | 2.98 | 57.58 |
| GPT4 | 64% | 4.50 | 69.09 | 4.73 | 63.00 | 5.12 | 65.62 | 2.89 | 58.38 |
| Reward Model | 64% | 4.43 | 70.03 | 4.76 | 64.92 | 5.50 | 71.88 | 2.91 | 59.18 |
| LossDiff-IRM | 64% | 4.82 | 76.36 | 5.02 | 68.85 | 5.47 | 74.38 | 3.19 | 64.83 |

**Pythia-1.4B (SLiC)**

| Methods | Dataset Ratio | UltraFeedback Single ↑ | WinRate ↑ | AlpacaEval Single ↑ | WinRate ↑ | Vicuna-Bench Single ↑ | WinRate ↑ | Arena-Hard Single ↑ | WinRate ↑ |
|---|---|---|---|---|---|---|---|---|---|
| SFT | | 3.50 | - | 3.65 | - | 4.20 | - | 2.37 | - |
| Full Data | 100% | 3.66 | 63.58 | 3.98 | 63.68 | 4.67 | 60.00 | 2.65 | 58.95 |
| Random | 52% | 3.81 | 66.18 | 3.96 | 63.99 | 4.34 | 56.25 | 2.66 | 60.26 |
| GPT4 | 52% | 3.84 | 69.24 | 4.12 | 65.55 | 4.69 | 63.75 | 2.68 | 59.80 |
| Reward Model | 52% | 3.82 | 66.57 | 4.04 | 69.20 | 4.67 | 65.62 | 2.67 | 61.95 |
| LossDiff-IRM | 52% | 4.14 | 74.25 | 4.39 | 72.69 | 4.88 | 71.25 | 2.85 | 67.76 |

**Pythia-410M (SLiC)**

| Methods | Dataset Ratio | UltraFeedback Single ↑ | WinRate ↑ | AlpacaEval Single ↑ | WinRate ↑ | Vicuna-Bench Single ↑ | WinRate ↑ | Arena-Hard Single ↑ | WinRate ↑ |
|---|---|---|---|---|---|---|---|---|---|
| SFT | | 2.56 | - | 2.47 | - | 3.15 | - | 1.91 | - |
| Full Data | 100% | 2.81 | 73.80 | 2.82 | 73.91 | 3.31 | 71.25 | 2.03 | 57.49 |
| Random | 56% | 2.67 | 69.23 | 2.69 | 70.68 | 3.09 | 63.12 | 2.14 | 59.77 |
| GPT4 | 56% | 2.80 | 72.75 | 2.85 | 74.53 | 3.23 | 75.62 | 2.15 | 60.07 |
| Reward Model | 56% | 2.87 | 77.41 | 2.94 | 78.39 | 3.50 | 75.00 | 2.15 | 60.11 |
| LossDiff-IRM | 56% | 3.07 | 80.08 | 3.09 | 84.39 | 3.83 | 79.36 | 2.21 | 62.40 |

Figure 3: **Ablation study on WinRate.** Comparison of LossDiff-IRM vs. its ablations that select data using only LossDiff ("w/ LossDiff") or only IRM ("w/ IRM"). *Top:* DPO training. *Bottom:* SLiC training. Single-Score results are provided in Appendix E.1 and Figure 12.

**Empirical evidence.** We quantify the overlap of two selected sets using the Overlap Coefficient $\text{Overlap}(A, B) = \frac{A \cap B}{\min\{|A|, |B|\}} \in [0, 1]$. As reported in Table 2, the combined selector LossDiff-IRM achieves a higher Overlap Coefficient with the exact TIF-selected set than using LossDiff or IRM alone across Qwen3-0.6B-Base and Llama-3.2-1B. This indicates that combining the two scoring functions yields a selection that more closely approximates the TIF while while being more computationally efficient at scale. As shown in the bottom of Table 1, LossDiff-IRM requires substantially less time and achieves higher throughput; for example, on Llama-3.2-1B it takes about 5 minutes versus roughly 10 hours for exact IF.

## 5 EXPERIMENTS

### 5.1 EXPERIMENTAL SETUP

**Setup.** Our experiments are conducted on diverse LLM families, including Llama-3.1-8B (Vavekanand & Sam, 2024), Qwen3-8B-Base (Yang et al., 2025), and the Pythia series (2.8B/1.4B/410M) (Biderman et al., 2023). Following prior work (Ko et al., 2024; Wu et al., 2024a), each pretrained LLM is first initialized with one epoch of supervised fine-tuning (SFT) on UltraChat-200k (Tunstall et al., 2024), which serves as the starting point for subsequent alignment. We adopt UltraFeedback-Binarized (Cui et al., 2023) as the alignment dataset, consistent with previous studies (Pattnaik et al., 2024; Gao et al., 2025; Ko et al., 2024). For validation set, we perform stratified sampling to select 20% of the alignment data according to the GPT-4 score margin between the chosen and rejected responses, and use the remaining 80% as training set. We employ

Table 4: **Performance comparisons of LossDiff-IRM with existing methods,** including Cur-riDPO (Pattnaik et al., 2024), $M_{AP}$ (Huang et al., 2025), and RS-DPO (Khaki et al., 2024).

| Training Data | UltraFeedback | | AlpacaEval | | Vicuna-Bench | | Arena-Hard | |
|---|---|---|---|---|---|---|---|---|
| | Single ↑ | WinRate ↑ | Single ↑ | WinRate ↑ | Single ↑ | WinRate ↑ | Single ↑ | WinRate ↑ |
| *Llama-3.1-8B (DPO)* | | | | | | | | |
| CurriDPO-GPT4 | 5.47 | 74.23 | 5.53 | 75.01 | 5.84 | 74.06 | 4.55 | 79.77 |
| CurriDPO-Reward Model | 5.51 | 74.62 | 5.54 | 74.29 | 5.59 | 74.06 | 4.62 | 79.49 |
| $M_{AP}$ | 6.04 | 79.99 | 6.21 | 80.88 | 6.34 | 74.69 | 5.08 | 85.30 |
| RS-DPO | 5.70 | 75.98 | 6.39 | 84.84 | 7.04 | 82.81 | 4.69 | 82.05 |
| LossDiff-IRM | 6.54 | 83.97 | 6.84 | 87.08 | 7.06 | 86.88 | 5.59 | 88.40 |
| *Qwen3-8B-Base (DPO)* | | | | | | | | |
| CurriDPO-GPT4 | 7.61 | 61.04 | 7.74 | 62.35 | 8.16 | 54.69 | 7.52 | 62.51 |
| CurriDPO-Reward Model | 7.60 | 59.62 | 7.84 | 61.96 | 8.20 | 61.58 | 7.55 | 63.97 |
| $M_{AP}$ | 7.95 | 67.84 | 8.31 | 71.11 | 8.62 | 66.87 | 7.72 | 66.55 |
| RS-DPO | 7.88 | 64.87 | 8.36 | 74.07 | 8.93 | 71.90 | 7.48 | 61.32 |
| LossDiff-IRM | 8.05 | 67.32 | 8.36 | 71.52 | 8.72 | 67.19 | 7.83 | 68.63 |

Figure 4: **Impact of noisy validation set.** Performance curves across different noise rates $r = \{0.0, 0.1, 0.2, 0.3, 0.4\}$; dashed lines denote performance of Full-Data training.

DPO (Rafailov et al., 2023) and SLiC (Zhao et al., 2023) as core preference optimization algorithms used in our experiments. More details of experimental settings are provided in Appendix D.

**Evaluation Setup.** We evaluate the aligned LLMs on open-ended generation tasks using both in-distribution (ID) and out-of-distribution (OOD) benchmarks. The ID evaluation is conducted on the UltraFeedback test split, while OOD evaluation includes AlpacaEval (Li et al., 2023), Vicuna-Bench (Chiang et al., 2023), and Arena-Hard (Li et al., 2024). We compare against full-data training (Full Data) and several preference data selection strategies commonly used in prior work (Pattnaik et al., 2024; Deng et al., 2025; Morimura et al., 2024): random sampling (Random), GPT-4 score filtering (GPT4), and external reward-model filtering (Reward Model). We also compare with several existing methods: curriculum learning-based CurriDPO (Pattnaik et al., 2024), margin-based selection $M_{AP}$ (Huang et al., 2025), and rejection sampling-based preference data generation method RS-DPO (Khaki et al., 2024). All methods are trained under the same setup for a fair comparison. We report two evaluation metrics via LLM-as-Judge (Zheng et al., 2023a): Single Score (Single) and Length-controlled Win Rate vs. SFT (WinRate). Further details are in Appendix D.3.

## 5.2 EXPERIMENTAL RESULTS

### 5.2.1 MAIN PERFORMANCE OF LOSS-IRM FOR PREFERENCE DATA SELECTION

**LossDiff-IRM achieves better performance.** Table 3 summarizes the performance of Loss-IRM and competitors with DPO and SLiC. Across diverse LLM families, benchmarks and metrics, it can be observed that LossDiff-IRM occupies more darker cells, and surpasses full-data training while using only about 50%–65% of the data, indicating stronger performance. Specifically, compared to full-data training, LossDiff-IRM achieves average WinRate improvements of +11.42%, +15.14%, +16.63%, +18.28% and +17.71% on Llama-3.1-8B, Qwen3-8B-Base, and Pythia-2.8B/1.4B/410M with DPO, respectively. Furthermore, we compute exact TIF on smaller Pythia-410M. Table 5 shows that training on data selected by exact TIF and by LossDiff-IRM achieves comparable performance. These results demonstrate the effectiveness and superiority of Loss-IRM in selecting real valuable preference pairs that are really beneficial for current model alignment training.

**LossDiff-IRM outperforms several data-centric baselines.** Table 4 reports the performance comparisons of LossDiff-IRM with several existing data-centric methods. It can be observed that our

Table 5: **Exact TIF vs. LossDiff-IRM.** Performance comparison of Pythia-410M trained on data selected by Exact TIF or LossDiff-IRM.

| Method | UltraFeedback | | AlpacaEval | | Vicuna-Bench | |
|---|---|---|---|---|---|---|
| | Single ↑ | WinRate ↑ | Single ↑ | WinRate ↑ | Single ↑ | WinRate ↑ |
| *Pythia-410M (DPO)* | | | | | | |
| Full Data | 2.81 | 75.25 | 2.77 | 73.51 | 3.10 | 69.37 |
| LossDiff-IRM | **3.30** | **86.14** | **3.30** | 85.16 | 3.80 | **85.62** |
| Exact TIF | 3.13 | 85.42 | 3.21 | 86.19 | **4.01** | 84.38 |

Figure 5: **Overlap Coefficient** of selections across five models.

Table 6: **Performance of training DPO and SLiC on Selected vs. Dropped set by LossDiff-IRM.** Comparison of training on Full Data, the LossDiff-IRM–selected data, and the dropped data.

| Training Data | UltraFeedback | | AlpacaEval | | Vicuna-Bench | | UltraFeedback | | AlpacaEval | | Vicuna-Bench | |
|---|---|---|---|---|---|---|---|---|---|---|---|---|
| | Single ↑ | WinRate ↑ | Single ↑ | WinRate ↑ | Single ↑ | WinRate ↑ | Single ↑ | WinRate ↑ | Single ↑ | WinRate ↑ | Single ↑ | WinRate ↑ |
| | *Llama-3.1-8B (DPO)* | | | | | | *Qwen3-8B-Base (DPO)* | | | | | |
| - Full Data | 5.77 | 77.61 | 5.87 | 78.41 | 6.04 | 73.75 | 7.64 | 61.41 | 7.92 | 63.85 | 8.21 | 62.14 |
| - w/ Selected Data | **6.54** | **83.97** | **6.84** | **87.08** | **7.06** | **86.88** | **8.05** | **67.32** | **8.36** | **71.52** | **8.72** | **67.19** |
| - w/ Dropped Data | 4.56 | 64.25 | 4.48 | 61.15 | 4.69 | 62.81 | 7.51 | 54.82 | 7.58 | 58.33 | 7.79 | 46.88 |
| | *Llama-3.1-8B (SLiC)* | | | | | | *Qwen3-8B-Base (SLiC)* | | | | | |
| - Full Data | 5.09 | 70.72 | 5.13 | 72.13 | 5.40 | 71.88 | 7.55 | 59.54 | 7.61 | 59.71 | 8.05 | 54.69 |
| - w/ Selected Data | **5.94** | **79.51** | **5.84** | **78.84** | **5.85** | **76.56** | **7.87** | **64.40** | **8.11** | **67.58** | **8.44** | **61.12** |
| - w/ Dropped Data | 4.22 | 60.08 | 4.33 | 59.37 | 4.89 | 62.81 | 7.37 | 56.67 | 7.33 | 55.79 | 8.21 | 56.72 |

LossDiff-IRM outperforms these competitors on the most cases. Specifically, the average improvement of WinRate achieves $+2.62\%$ over the best baselines across two models, especially achieving improvements of $+6.45\%$ on average on Llama-3.8-8B. These results suggest the effectiveness of our IF-driven analysis and LossDiff-IRM data selection strategy. Additionally, compared to CurriDPO relied on external signals (GPT4 or reward model scores), $M_{AP}$ and RS-DPO, which adopt implicit reward margins or generate model-specific preference pairs using the SFT model and thus partially incorporate the model's perspective, emerge as strong competitors. This aligns with the underlying philosophy of our idea: valuable preference data are model-dependent, rather than relying on external heuristics. The cost analysis between LossDiff-IRM and RS-DPO is provided in Appendix E.5.

**Preference data quality is model-dependent.** We observe that LossDiff-IRM outperforms selection based on Random, GPT4 score or external reward model, with only a few single-case exceptions, e.g., Qwen3-8B-Base on Vicuna-Bench under SLiC. Under DPO, relative to the second-best result, LossDiff-IRM delivers average WinRate improvements of $+4.07\%$, $+3.84\%$, $+8.28\%$, $+10.29\%$ and $+8.13\%$ on Llama-3.1-8B, Qwen3-8B-Base, and Pythia-2.8B/1.4B/410M, respectively. Notably, GPT4 score selection can even reduce performance, such as on Qwen3-8B-Base with DPO. Moreover, Figure 5 shows that the Overlap Coefficient between selections varies across models; pairs within the same family (e.g., Pythia) exhibit higher overlap than cross-family pairs. Altogether, these findings indicate that preference data quality is inherently model dependent, and static or model-agnostic selectors may benefit one model yet harm another.

**LossDiff-IRM is compatible to different preference optimization methods.** For both DPO and SLiC, LossDiff-IRM selection yields consistent and often substantial gains across diverse LLM families and architectures, indicating the generality of the LossDiff-IRM method. This compatibility stems from that the derivation and analysis of LossDiff-IRM criterion do not involve any strong assumption about certain preference optimization algorithm, so that diverse methods can be initiated under our LossDiff-IRM criterion, which serves as a plug-and-play preference data selection step.

### 5.2.2 FURTHER ANALYSIS

**Combining LossDiff and IRM outperforms either alone.** Figure 3 reports WinRate for LossDiff-IRM and its two ablations: "w/ LossDiff" and "w/ IRM", which select pairs using only LossDiff or only IRM, respectively. Across both DPO and SLiC, the combined scoring function LossDiff-IRM performs better than either single variant. This matches our intent: combining the two scoring functions to offset specific errors each incurs when used alone to approximate TIF in data selection.

**Dropped data is low-value for the current model alignment.** We train DPO and SLiC using only the subset that LossDiff-IRM drops and compare against Full Data and the LossDiff-IRM–selected

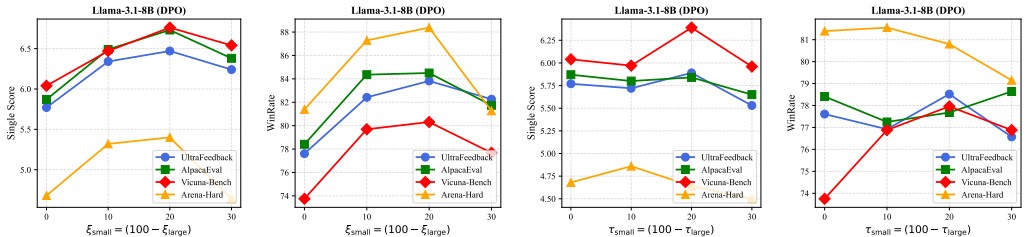

Figure 6: **Analysis of percentile thresholds $\xi_{\text{small}}, \xi_{\text{large}}, \tau_{\text{small}}, \tau_{\text{large}}$ of Llama-3.1-8B.** We vary $\xi_{\text{small}} = (100 - \xi_{\text{large}})$ and $\tau_{\text{small}} = (100 - \tau_{\text{large}})$ with in $\{0, 10, 20, 30\}$. Further analysis of Qwen3-8B-Base is illustrated in Figure 14 and Appendix E.6.

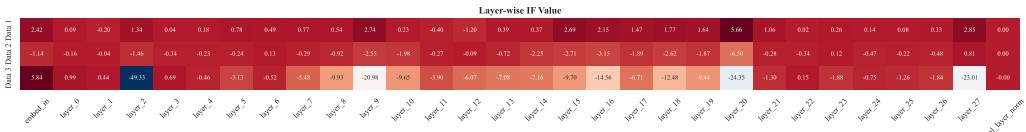

Figure 7: **Visualization of layer-wise IF value computed on Qwen3-0.6B-Base.**

subset. Table 6 shows that training "w/ Dropped Data" yields the worst performance; in particular, it reduces average WinRate by $-12.59\%$ relative to full-data training across two models and both alignment methods, and even drops to 46.88% on Vicuna-Bench with Qwen3-8B-Base (SLiC). By contrast, "w/ Selected Data" consistently improves alignment, indicating that LossDiff-IRM filters out low-value pairs and favors data that really benefits the current model alignment learning.

**LossDiff-IRM exhibits a certain robustness under validation-set noise.** We corrupt the validation set by flipping the chosen/rejected labels at rates $r \in \{0.1, 0.2, 0.3, 0.4\}$ and then re-run selection. Figure 4 shows the expected downward trend in performance as $r$ increases, since LossDiff-IRM (and TIF) uses the validation set to orient the score. Surprisingly, in many cases the selection upon noisy validation set still exceeds full-data training (dashed lines) under noisy validation set. We attribute this to the fact that LossDiff-IRM tends to medium-value data that are less sensitive to validation noise. These results suggest the robustness of LossDiff-IRM to noisy validation set.

**Analysis of percentile thresholds $\xi_{\text{small}}, \xi_{\text{large}}, \tau_{\text{small}}, \tau_{\text{large}}$.** The percentile thresholds of LossDiff-IRM are tuned as hyperparameters. We vary $\xi_{\text{small}} = (100 - \xi_{\text{large}})$ and $\tau_{\text{small}} = (100 - \tau_{\text{large}})$ with in $\{0, 10, 20, 30\}$. Figure 6 illustrates the performance curves varying with different thresholds for Llama-3.1-8B (DPO) across four benchmarks. It can be observed a rough trend: performance first improves and then degrades as the thresholds become stricter, that is, as more data are filtered out. This observation is expected, because initially, the threshold helps remove low-quality data; however, beyond a certain point, the filtering starts to exclude informative and high-quality data, leading to performance degradation. More analysis on Qwen3-8B-Base is provided in Appendix E.6.

**IF values are not confined to specific layers.** Figure 7 shows that IF may concentrate in certain layers or spread across all layers, implying that layer-wise computation alone cannot reliably approximate the exact IF and motivating the need for proxies such as our LossDiff–IRM. More visualization of IF computed on other LLMs are provided in Appendix E.2.

# 6 CONCLUSION

In this work, we propose the Truncated Influence Function (TIF) as a principled lens to analyze preference data quality in LLM alignment. Unlike prior approaches that treat data quality as an inherent property of the data, our analysis adopts a model-side perspective and reveals that medium-IF pairs, rather than small- or large-IF ones, provide the most effective training signal. To make TIF practical at scale, we further introduce the LossDiff–IRM approximation, which closely matches TIF while being far more efficient. Experiments demonstrate that LossDiff–IRM enables a "less is more" effect, where using fewer but higher-quality preference pairs yields better alignment performance.

ACKNOWLEDGMENTS

ZZZ, QZW, JNZ and BH were supported by RGC General Research Fund No. 12200725, RIKEN Collaborative Research Fund, and HKBU CSD Departmental Incentive Scheme. MS was supported by JST ASPIRE Grant Number JPMJAP25B1. JCY is supported by National Natural Science Foundation of China (No. 62306178) and STCSM (No. 22DZ2229005).

ETHICS STATEMENT

This study adheres to the Code of Ethics. It relies only on publicly available models and datasets, without the use of sensitive or privacy data, and poses no identifiable risks concerning privacy, security, or fairness. The work is conducted purely for scientific purposes, and no conflicts of interest are involved.

REPRODUCIBILITY STATEMENT

We are committed to ensure the reproducibility of our proposed method. The detailed descriptions of our approach and experimental settings are all provided in this paper. The corresponding source code is released at https://github.com/tmlr-group/TIF_LossDiff-IRM. Both backbone models and datasets used in our work are publicly available. Furthermore, all parameters, hyper-parameters, and procedural steps required to reproduce our results are thoroughly recorded in the Experimental Details section and corresponding Appendix. We believe that these components provide the community with details necessary to verify and reproduce our work.

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

## LLM USAGE STATEMENT

Here we clarify how Large Language Models (LLMs) are used in this work. In preparing the manuscript, LLMs served only as a writing assistants for writing improvements and were not involved in research ideation or the generation of core content. For research methodology, LLM is a core component of our proposed method. Specifically, we utilize the Llama-3.1-8B, Qwen3-8B-Base and Pythia series as our backbone models to validate our proposed method.

## A  BACKGROUND AND RELATED WORK

### A.1  SUPPLEMENT TO PRELIMINARY

**Reinforcement Learning with Human Feedback (RLHF).** RLHF aims to align the given policy model $\pi_\theta$ with human preference by optimizing the model to maximize the expected reward value obtained from the reward model (Zhang et al., 2025). The reward model is trained on a preference dataset, quantifying the human preference into scalar values. Typically, the Bradley-Terry (BT) model (Bradley & Terry, 1952) is used to estimate the probability distribution that a chosen response $y_w$ is preferred over a rejected one $y_l$ as follows:

$$p(y_w \succ y_l|x) = \frac{\exp\left(r(x, y_w)\right)}{\exp\left(r(x, y_w)\right) + \exp\left(r(x, y_l)\right)} = \sigma(r(x, y_w) - r(x, y_l)), \quad (10)$$

where $r(x, y)$ denotes the latent reward function, and $\sigma(\cdot)$ is the Sigmoid function. Due to unobservability of the reward function, the traditional RLHF typically follows a two-stage pipeline: a reward model $r_\phi(x, y)$ is pretrained on preference data; then, the policy model $\pi_\theta$ is updated based on the pretrained reward model using reinforcement learning (RL) algorithm like PPO (Schulman et al., 2017). The overall objective of RLHF can be formulated as follows:

$$\mathcal{L}_{\text{RLHF}}(\theta; \mathcal{D}) = -\mathbb{E}_{x \sim \mathcal{D}, y \sim \pi_\theta(\cdot|x)}\left[r_\phi(x, y)\right] + \beta \mathbb{D}_{\text{KL}}[\pi_\theta(y|x)||\pi_{\text{ref}}(y|x)], \quad (11)$$

where $\mathbb{D}_{\text{KL}}[\cdot||\cdot]$ is the KL-divergence regulation term that constrains the policy model $\pi_\theta$ to optimize within the surrounding landscape of the reference model $\pi_{\text{ref}}$, avoiding policy collapse or training instablity during alignment, and $\beta$ is a hyperparameter to control trade-off between reward maximization and KL penalty. The reference model is often the supervised fine-tuned (SFT) model (Zhang et al., 2023) used to initialize the policy LLM.

### A.2  RELATED WORK

**LLM Preference Alignment.** LLM has achieved remarkable success across a wide range of tasks (Li et al., 2026; Chen et al., 2026). LLM preference alignment aims to steer LLM behaviors toward responses that better reflect human preferences and values (Wang et al., 2024). A common approach is RLHF that trains a reward model to provide reward signals for RL training (Bai et al., 2022b; Zhou et al., 2025a;b). Though effective, the two-stage RLHF pipeline is complex and resource-intensive, suffering from issues like reward hacking (Miao et al., 2024) and unstable optimization (Zheng et al., 2023b; Zhang et al., 2026). To address these limitations, recent studies (Rafailov et al., 2023; Azar et al., 2024; Zhao et al., 2023; Meng et al., 2024; Ethayarajh et al., 2024) have proposed multiple alignment learning objectives that bypass explicit reward modeling, offering a simpler formulation for direct LLM preference alignment, such as DPO (Rafailov et al., 2023), IPO (Azar et al., 2024), SLiC (Zhao et al., 2023; Liu et al., 2024), SimPO (Meng et al., 2024), KTO (Ethayarajh et al., 2024), CPO (Xu et al., 2024) and more (Wu et al., 2024b;a; Liu et al., 2024). Among them, DPO serves as a milestone work that derives the close-form expression of the optimal policy and substitutes it into the Bradley–Terry (BT) model, effectively hiding reward learning within the policy optimization process. SLiC adopts a hinge loss to enlarge the margin between the chosen and rejected responses. SimPO simplifies the DPO formulation and achieves reference model-free optimization. KTO employ prospect theory to directly maximizes the utility of generations rather than log-likelihood of preferences. Overall, while many efforts have been devoted to improving alignment algorithms, relatively little attention (Deng et al., 2025; Gao et al., 2025) has been given to understanding the preference data quality, especially from model perspective.

**Data Selection for LLMs.** Data selection (Albalak et al., 2024) aims to filter out low-quality or noisy data (Zhou et al., 2024) and retain high-quality data for better training and generalization,

which plays a pivotal role in enhancing the performance and efficiency of LLMs across difference training stages, including pretraining (Penedo et al., 2023; Tang et al., 2024), supervised fine-tuning (SFT) (Pang et al., 2025; Qin et al., 2024), and preference alignment (Shen et al., 2024; Gao et al., 2025). Most existing approaches assess data from multiple considerations, such as importance (Xie et al., 2023) and diversity (Zhang et al., 2024a), using a variety of metrics such as text length (Nagatsuka et al., 2023), perplexity (Kong et al., 2024), text embeddings (Saranathan et al., 2024), or external signals from humans (Wu et al., 2023) or ChatGPT (Chen et al., 2024). We notice that most studies for LLM data selection primarily focus on the first two stages, i.e., pretraining and SFT, whereas the selection of preference data for alignment has received comparatively limited exploration. Existing studies (Pattnaik et al., 2024; Morimura et al., 2024; Deng et al., 2025) often employ off-the-shelf LLMs or pretrained reward models to pre-process preference data. For example, CurriDPO (Pattnaik et al., 2024) leverages the GPT-4 or reward model scores to organize curriculum learning. fDPO (Morimura et al., 2024) integrates a reward model into the DPO training process, filtering preference data based on reward scores. Muldrew et al. (2024) and Shen et al. (2025) incorporate active learning into alignment to improve data quality and annotation efficiency. Overall, these methods share a common trait: they rely on external signals–whether from powerful GPT models, pretrained reward models, or active human labeling–and thereby treat data quality as an inherent property of the data itself, while overlooking the role of the model and training objectives. Motivated by this limitation, we examine preference data quality from the model's perspective to better understand which data are truly valuable for alignment.

# B  FORMAL DERIVATION

## B.1  FORMULATIONS OF SLiC LOSS

**Definition of SLiC Loss.** Unlike DPO loss in Eq. (1), which optimizes a log-ratio difference between chosen and rejected responses, SLiC (Zhao et al., 2023; Liu et al., 2024) minimizes a hinge loss on normalized log-likelihood, formulated as follows:

$$\mathcal{L}_{\text{SLiC}} = \mathbb{E}_{(x,y_w,y_l)\sim\mathcal{D}} \left[ \max\left(0, 1 - \left(\beta \log \frac{\pi_\theta(y_w|x)}{\pi_{\text{ref}}(y_w|x)} - \beta \log \frac{\pi_\theta(y_l|x)}{\pi_{\text{ref}}(y_l|x)}\right)\right) \right]. \tag{12}$$

Intuitively, SLiC loss enforces at least a fixed margin between the chosen and rejected responses. When the margin is satisfied, the loss becomes zero, making the optimization focus on those pairs that violate the margin constraint.

**Instantiation of SLiC Loss for Influence Score.** By taking SLiC loss into the influence function formulation in Eq. (5), the SLiC-based influence function is derived as follows:

$$\text{IF}_{\text{SLiC}}(d; \pi_\theta; \mathcal{D}_{\text{val}}) := \beta^2 \mathbb{I}_{\text{hinge}}(d) \left\langle \underbrace{\frac{1}{|\mathcal{D}_{\text{val}}|} \sum_i \mathbb{I}_{\text{hinge}}\left(d_{\text{val}}^{(i)}\right)\left(g_w^{(i)} - g_l^{(i)}\right)}_{\substack{\text{preference generalization direction} \\ \text{w.r.t. validation set}}}, \underbrace{g_w - g_l}_{\substack{\text{current preference} \\ \text{pair direction}}} \right\rangle, \tag{13}$$

where

$$\mathbb{I}_{\text{hinge}}(d) = \mathbb{I}[1 - \Delta_\theta > 0] \quad \text{and} \quad \Delta_\theta = \beta \log\left[\frac{\pi_\theta(y_w|x)}{\pi_{\text{ref}}(y_w|x)} - \log \frac{\pi_\theta(y_l|x)}{\pi_{\text{ref}}(y_l|x)}\right]. \tag{14}$$

Both DPO and SLiC assign higher IF values to preference pairs whose gradient difference direction (i.e., $g_w - g_l$) is consistent with that of the validation preferences, and negative influence scores when they oppose. In DPO, the IF is further scaled by $1 - \sigma(\Delta_\theta)$, assigning near-zero scores to pairs with large $\Delta_\theta$, i.e., those already well learned by the model. Similarly, SLiC sets the IF to exactly zero for margin-satisfied pairs ($\Delta_\theta > 1$). These well learned pairs naturally fall into the medium-IF region, and are also considered as high-quality data that promote preference generalization.

## B.2  RELATIONSHIP BETWEEN INFLUENCE INFLUNCTION AND LOSSDIFF

To analyze the positive correlation between influence function defined in Eq. (4) and LossDiff defined in Eq. (8), we provide formal justification as follows:

**Lemma B.1** (Loss Difference (LossDiff) Correlates with Influence Function (IF)). *Assume that the validation-aligned model $\pi_{\theta_{val}}$ is obtained by performing a single gradient descent step from the current model $\pi_\theta$ on the loss, i.e., $\theta_{val} = \theta - \eta \nabla_\theta \mathcal{L}(\theta; \mathcal{D}_{val})$, where $\eta$ is the learning rate. For a give training preference pair $d$, the loss difference and the influence function are positively correlated:*

$$LossDiff(d; \pi_\theta, \pi_{\theta_{val}}) := \ell(\theta; d) - \ell(\theta_{val}; d) \propto I(d; \pi_\theta; \mathcal{D}_{val}), \qquad (15)$$

*where $I(d; \pi_\theta; \mathcal{D}_{val})$ is the influence function defined in Eq. (5).*

*Formal Derivation.* By applying a first-order Taylor expansion of the loss $\ell(\theta; d)$ at $\theta$:, we get:

$$\ell(\theta_{\text{val}}; d) \approx \ell(\theta; d) + \nabla_\theta \ell(\theta; d)^\top (\theta_{\text{val}} - \theta). \qquad (16)$$

Then, rearranging terms gets:

$$\ell(\theta; d) - \ell(\theta_{\text{val}}; d) \approx -\nabla_\theta \ell(\theta; d)^\top (\theta_{\text{val}} - \theta). \qquad (17)$$

Now assume $\theta_{\text{val}} = \theta - \eta \nabla_\theta \mathcal{L}(\theta; \mathcal{D}_{\text{val}})$, where $\eta > 0$ is the learning rate. Substituting this into the equation yields:

$$\ell(\theta; d) - \ell(\theta_{\text{val}}; d) \approx \eta \cdot \nabla_\theta \ell(\theta; d)^\top \nabla_\theta \mathcal{L}(\theta; \mathcal{D}_{\text{val}}) \qquad (18)$$

$$\propto \nabla_\theta \ell(\theta; d)^\top \nabla_\theta \mathcal{L}(\theta; \mathcal{D}_{\text{val}}) \qquad (19)$$

$$= I(d; \pi_\theta; \mathcal{D}_{\text{val}}). \qquad (20)$$

Hence, the LossDiff defined in Eq. (8) is positively correlated to the IF defined in Eq. (4). $\qquad \square$

The formal justification assumes that the validation-aligned model $\pi_{\theta_{\text{val}}}$ is obtained by performing a single gradient descent step from the current model $\pi_\theta$ on the alignment loss computed over the validation set. While this assumption does not precisely reflect the actual training procedure, it serves as a reasonable local approximation. In preference alignment methods, they all explicitly or implicitly have a KL divergence term in preference optimization objectives (Bai et al., 2022b; Rafailov et al., 2023). The KL regularization term constrains both the $\pi_\theta$ and the $\pi_{\theta_{\text{val}}}$ to remain close to the reference model in parameter space. Since both $\theta$ and $\theta_{\text{val}}$ lie in a small neighborhood around the same reference model, it is reasonable to assume that $\theta_{\text{val}}$ can be approximated by a single gradient descent step from $\theta$ on the preference loss. This justifies the assumption used in Lemma B.1.

### B.3 Formal Analysis of the Reliability of Medium-IF Data Selection

We demonstrate that the reliability of selecting high-IF data decreases as the quality of the validation data decreases. We provide the derivation pipeline as follows:

Suppose the validation dataset is corrupted w.p. $\eta$. Then, the expected validation gradient $v$ consists of the clean ($c$) and noisy ($n$) parts, following $g_v = (1 - \eta)g_c + \eta g_n$. For a training sample $i$ with the gradient $g_i$, we have its influence weight as

$$\omega_i = -g_v^\top g_i = -(1 - \eta)g_c^\top g_i - \eta g_n^\top g_i, \qquad (21)$$

which can be further written as $\omega_i = \omega_i^* + \delta_i$, with $\omega_i^*$ the clean weight and $\delta_i = -\eta(g_n - g_c)^\top g_i$ the bias term. Assuming that $g_n - g_c$ is independent on $g_i$, with zero mean and isotropic variance $\sigma_n^2 I$, then we have $\text{Var}(\delta_i) = \eta^2 \sigma_n^2 \|g_i\|^2$ and $\text{Var}(\omega_i) = \sigma_w^2 + \tau^2$, with $\sigma_w^2 = \text{Var}(\omega_i)$ and $\tau^2 = \text{Var}(\delta_i)$. Accordingly, we have the expected correlation between $\omega^*$ and $\omega$ as

$$\rho = \frac{\text{Cov}(\omega_i^*, \omega_i)}{\sqrt{\text{Var}(\omega_i^*)\text{Var}(\omega_i)}} = \sqrt{\sigma_w^2 / \sigma_w^2 + \tau^2}. \qquad (22)$$

As observed, when the corruption rate $\eta$ increases, the correlation $\rho$ between the true and observed scores decreases, leading to a lower probability $p(i \in \mathcal{T}_{\text{true}}(k) | i \in \mathcal{T}_{\text{obs}}(k))$. Hence, the top-$k$ sampling strategy becomes less reliable under high noise levels. Similar derivations hold for bottom-$k$ sampling, making middle-$k$ sampling tend to be more reliable.

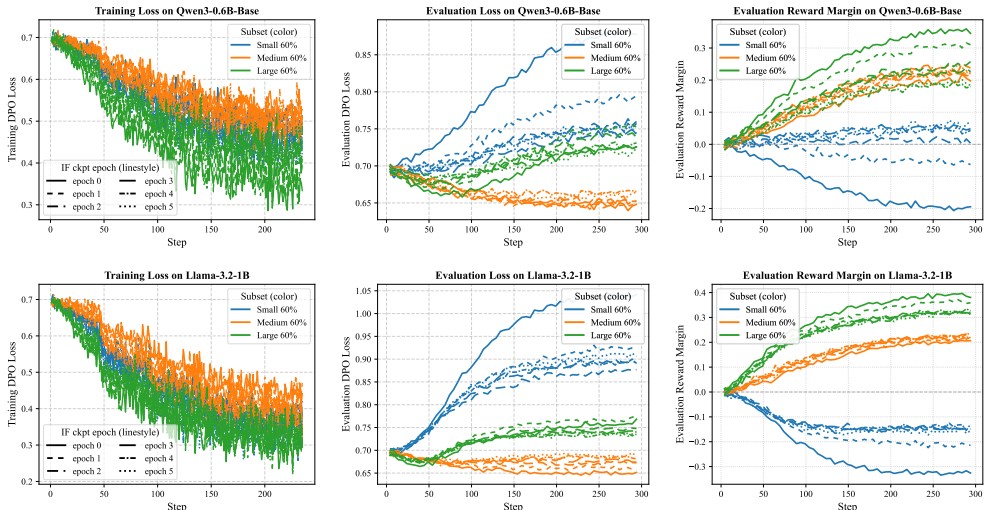

Figure 8: **Analysis of IF-based data partition with overlapping splits on Qwen3-0.6B-Base and Llama-3.2-1B.** *From Left to Right:* Trainin DPO loss, evaluation DPO loss and evaluation reward margin. Subsets of {small, medium, large}-60% are denoted by {blue, orange, green}, respectively, while different line styles indicate IF values computed using different epoch checkpoints. Previous analysis refers to Section 3.2 and Figure 1.

Table 7: **Further Overlap Coefficient analysis** on epoch-{3,4,5} DPO checkpoints to assess the overlap between two selected sets.

| Overlap Coefficient | LossDiff vs. IF | | | IRM vs. IF | | | LossDiff-IRM vs. IF | | |
|---|---|---|---|---|---|---|---|---|---|
| Models | Epoch 3 ckpt | Epoch 4 ckpt | Epoch 5 ckpt | Epoch 3 ckpt | Epoch 4 ckpt | Epoch 5 ckpt | Epoch 3 ckpt | Epoch 4 ckpt | Epoch 5 ckpt |
| Qwen3-0.6B-Base | 0.6593 | 0.6343 | 0.6340 | 0.6153 | 0.5983 | 0.6000 | **0.6917** | **0.6770** | **0.6747** |
| Llama-3.2-1B | 0.6319 | 0.6299 | 0.6292 | 0.5645 | 0.5512 | 0.5499 | **0.6687** | **0.6643** | **0.6673** |

## C  ADDITIONAL ANALYSIS

### C.1  MORE ANALYSIS OF IF ON QWEN3-0.6B-BASE AND LLAMA-3.2-1B

As a supplement to Sec 3.2, we investigate whether introducing more medium-IF data into the small-IF and large-IF subsets can mitigate the adverse effects of extreme IF values. To this end, we adopt overlapping splits, dividing the data into large-60%, medium-60%, and small-60% subsets based on their IF scores, where the small-60% and large-60% subsets partially overlap with the medium-60% subset. Figure 8 shows the training dynamics of such overlapping data partition on Qwen3-0.6B-Base and Llama-3.2-1B. We observe similar phenomena shown as non-overlapping splits in Figure 1: small-IF data continues to be uninformative and large-IF data still drives overfitting. This suggests that extremely small- or large-IF pairs are detrimental, as even introducing smoother overlaps fails to mitigate their negative effects.

### C.2  MORE ANALYSIS OF CORRELATION

As a supplement to Figure 2 and Table 2, which analyze the epoch-1,2 checkpoints of Qwen3-0.6B-Base and Llama-3.2-1B, Figure 9 and Table 7 present the results for the remaining epoch-3,4,5 checkpoints. We observe that as training progresses, the correlation between IRM and IF decreases more sharply than that between LossDiff and IF. For example, at epoch-5 of Llama-3.2-1B, the Pearson coefficient between IRM and IF drops to 0.0636. Nevertheless, the Overlap Coefficients of LossDiff–IRM remain consistently above 0.65, indicating a relatively high level of agreement with the exact IF-based selection and exceeding those of LossDiff or IRM alone. This supports our design choice: the errors of LossDiff and IRM in approximating IF are diverse and may partially offset each other, so combining them yields a selection that more closely matches the exact IF-based criterion.

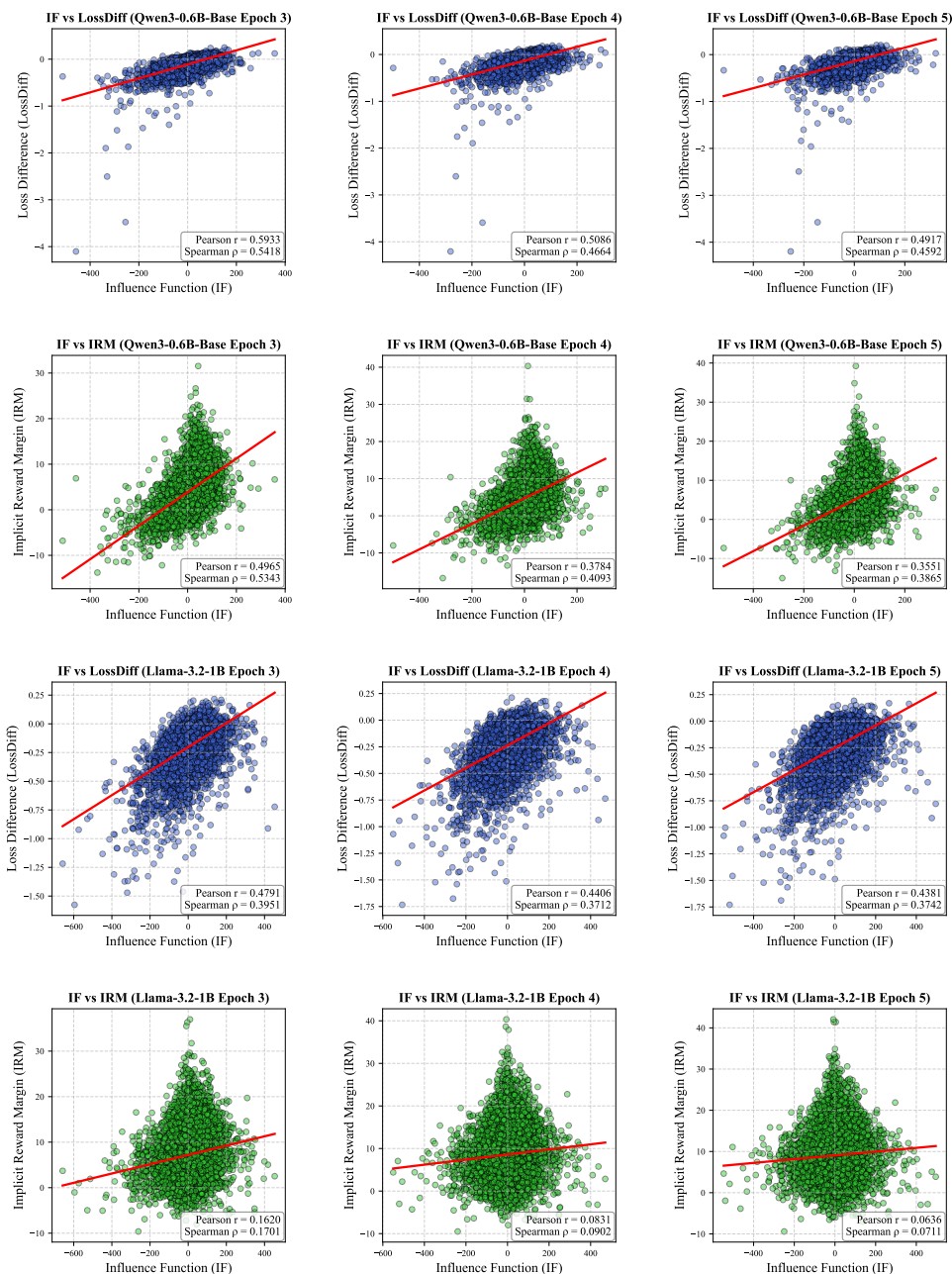

Figure 9: **Further correlation analysis on epoch-{3,4,5} checkpoints of Qwen3-0.6B-Base and Llama-3.2-1B.** *First Row:* correlation between LossDiff and IF on Qwen3-0.6B-Base. *Second Row:* correlation between IRM and IF on Qwen3-0.6B-Base. *Third Row:* correlation between LossDiff and IF on Llama-3.2-1B. *Fourth Row:* correlation between IRM and IF on Llama-3.2-1B.

# D   MORE EXPERIMENTAL DETAILS

## D.1   DATASET DETAILS

The details of the datasets used in this work is introduced as follows:

Table 8: **Statistics of the datasets** used in this work.

| Dataset | Purpose | # Instances | | | Unit |
|---|---|---|---|---|---|
| | | Train | Val | Eval | |
| UltraChat-200K | SFT | 207,865 | - | 23,110 | Dialogues |
| UltraFeedback-Binarized | Pref. Train | 48,908 | 12,227 | - | Pairs |
| UltraFeedback Test set | Evaluation | - | - | 1,000 | Pairs |
| AlpacaEval | Evaluation | - | - | 805 | Prompts |
| Vicuna-Bench | Evaluation | - | - | 80 | Prompts |
| Arena-Hard | Evaluation | - | - | 500 | Prompts |

- **UltraChat-200K**[1] (**Tunstall et al., 2024**): This is a heavily filtered subset version of Ultra-Chat (Ding et al., 2023), which was originally used for training ZePhyr-7B-$\beta$ model. The filtering process removes dialogues containing grammatical errors or assistant responses with phrases such as "I do not have emotions" or "I don't have opinions." After filtering, training set of UltraChat-200K contains 207,865 multi-turn dialogues generated by ChatGPT, covering a wide range of topics. Currently, it is widely adopted for supervised fine-tuning (SFT) of LLMs in research community (Ko et al., 2024; Zhang et al., 2024b). We also use this dataset to perform SFT on LLMs as preparation for subsequent preference alignment.

- **UltraFeedback-Binarized**[2] (**Cui et al., 2023**): This is a preprocessed pairwise version of the UltraFeedback[3] dataset, designed for LLM preference alignment. The dataset contains 64k prompts collected from diverse sources. For each prompt, four responses are generated by different LLMs and then evaluated by GPT-4 along four axes: instruction-following, truthfulness, honesty, and helpfulness. To construct preference pairs of the UltraFeedback-Binarized, the response with the highest overall score is selected as the "chosen" response, while one of the remaining three is randomly selected as the "rejected" response. We perform stratified sampling based on the GPT-4 score difference between chosen and rejected responses, holding out 20% of the training split as a validation set, using the remaining 80% as our full training set in this work. The resulting dataset contains 48,908 training pairs and 12,227 validation pairs.

- **UltraFeedback Test Set** (**Cui et al., 2023**): This test set is provided alongside the UltraFeedback-Binarized dataset and constructed using the same preprocessing procedure. It contains 2,000 high-quality preference pairs. To reduce the cost of LLM-based evaluation in our experiments, we randomly sample 1,000 pairs from this set to form the UltraFeedback evaluation dataset used in this paper. To reduce the cost of LLM-as-a-Judge (Zheng et al., 2023a) evaluation, we randomly sample 1,000 pairs from this set to construct the UltraFeedback benchmark used in this work.

- **AlpacaEval**[4] (**Li et al., 2023**): This is a lightly modified version of the AlpacaFarm (Dubois et al., 2023) evaluation set, containing 805 challenging prompts spanning a wide range of topics. Following previous studies (Ko et al., 2024; Gao et al., 2025), we employ AlpacaEval to evaluate the instruction-following capability of the trained LLMs in this paper.

- **Vicuna-Bench** (**Chiang et al., 2023**): This is a dataset containing 80 diverse questions originally used to evaluate the Vicuna series of LLMs. Following previous work (Pattnaik et al., 2024; Ko et al., 2024), we adopt it to evaluate the open-ended question-answering ability of the model.

- **Area-Hard**[5] (**Li et al., 2024**): This is a benchmark containing 500 challenging prompts curated by BenchBuilder, which utilizes LLM judges to estimate human preferences.

The statistics information of above datasets used in this paper is summarized in Table 8.

---

[1]https://huggingface.co/datasets/HuggingFaceH4/ultrachat_200k

[2]https://huggingface.co/datasets/HuggingFaceH4/ultrafeedback_binarized

[3]https://huggingface.co/datasets/openbmb/UltraFeedback

[4]https://huggingface.co/datasets/tatsu-lab/alpaca_eval

[5]https://huggingface.co/datasets/lmarena-ai/arena-hard-auto

Table 9: **Training Setup Details.**

| Stage | Hyperparameter | Llama-3.1-8B | Qwen3-8B-Base | Pythia-410M | Pythia-1.4B | Pythia-2.8B |
|---|---|---|---|---|---|---|
| SFT | Learning rate | | | 2e-5 | | |
| | Optimizer | | | AdamW | | |
| | Scheduler | | | Cosine | | |
| | # Epoch | | | 1 | | |
| | Batch Size | 8 | 8 | 64 | 32 | 8 |
| | Gradient accumulations | 8 | 8 | 1 | 2 | 8 |
| Alignment | Learning rate | 2e-4 | | | 5e-7 | |
| | Optimizer | | | AdamW | | |
| | Scheduler | | | Linear | | |
| | # Epoch | | | 2 | | |
| | Batch Size | 16 | 16 | 32 | 8 | 2 |
| | Gradient accumulations | 1 | 1 | 1 | 4 | 16 |
| | $\beta$ | | | 0.1 (DPO) / 0.1 (SLiC) | | |
| | $r_{\mathrm{LoRA}}$ | 32 | 32 | | - | |
| | $\alpha_{\mathrm{LoRA}}$ | 32 | 32 | | - | |
| | $\mathrm{drop\_out}_{\mathrm{LoRA}}$ | 32 | 32 | | - | |

Please act as an impartial judge and evaluate the quality of the response provided by an AI assistant to the user question displayed below. Your evaluation should consider factors such as the helpfulness, relevance, accuracy, depth, creativity, and level of detail of the response. Begin your evaluation by providing a short explanation. Be as objective as possible. After providing your explanation, please finally rate the response on a scale of 1 to 10 by strictly following this format: "[[rating]]", for example: "Rating: [[5]]".

[Question]
{question}

[The Start of Assistant's Answer]
{response}
[The End of Assistant's Answer]

Figure 10: **Pointwise single-answer grading prompt to compute metric of "Single".**

## D.2 TRAINING DETAILS

We conduct experiments using various LLM families, including Llama-3.1-8B (Vavekanand & Sam, 2024), Qwen3-8B-Base (Yang et al., 2025) and the Pythia series (Pythia-2.8B/1.4B/410M) (Biderman et al., 2023). Due to GPU limit, Llama-3.1-8B and Qwen3-8B-Base are trained using LoRA with $r_{\mathrm{LoRA}} = 32, \alpha_{\mathrm{LoRA}} = 32$ and $\mathrm{drop\_out} = 0.05$, whereas the Pythia models are trained in a full-parameter setting. As described in the previous section, we first perform one epoch of SFT on each model based on UltraChat-200K dataset to initialize the alignment learning.

We then apply two preference alignment algorithms, i.e, DPO and SLiC, to validate the effectiveness of our proposed LossDiff-IRM. For both DPO and SLiC, we train for two epochs with the AdamW optimizer and a linear learning rate scheduler. The batch sizes and gradient accumulation steps are set to {16, 16, 32, 8, 2} and {1, 1, 1, 4, 16} for the five models, respectively. The hyperparameter of KL pernalty $\beta$ in DPO and SLiC is set 0.1 for both DPO and SLiC following previous work (Ko et al., 2024). All experiments are conducted using bfloat16 dtype in our experiments. All experiments are conducted on two NVIDIA H100-80GB GPU using the Hugging Face TRL[6] library. All training hyperparameters refer to Table 9

---

[6]https://huggingface.co/docs/trl/index

Please act as an impartial judge and evaluate the quality of the responses provided by two AI assistants to the user question displayed below. You should choose the assistant that follows the user's instructions and answers the user's question better. Your evaluation should consider factors such as the helpfulness, relevance, accuracy, depth, creativity, and level of detail of their responses. Begin your evaluation by comparing the two responses and provide a short explanation. Avoid any position biases and ensure that the order in which the responses were presented does not influence your decision. Do not allow the length of the responses to influence your evaluation. Do not favor certain names of the assistants. Be as objective as possible. After providing your explanation, output your final verdict by strictly following this format: "[[A]]" if assistant A is better, "[[B]]" if assistant B is better, and "[[C]]" for a tie.

[User Question]
{question}

[The Start of Assistant A's Answer]
{response_a}
[The End of Assistant A's Answer]

[The Start of Assistant B's Answer]
{response_b}
[The End of Assistant B's Answer]

Figure 11: **Pairwise comparison prompt to compute metric of "WinRate vs. SFT".**

## D.3 EVALUATION DETAILS

To evaluate the aligned LLMs, we use the vllm[7] library to accelerate inference and generate responses with sampling temperature set to 1.0, top-$p$ of 0.95, and a maximum generation length of 512 tokens to control the generation length. For LLM-as-a-judge evaluation, we adopt GLM-4-Plus[8] Pythia experiments, while for Llama-3.1-8B and Qwen3-8B-Base we adopt the open-source Qwen3-32B to reduce API token costs. We adopt two types of prompting strategies: (1) a pointwise single-answer grading prompt, which yields the Single Score, and (2) a pairwise comparison prompt, which produces the Win Rate vs. SFT. The temperature of the judge is set to 0.0 for both two metrics. The prompt templates for both evaluation metrics are shown in Fig. 10 and Fig. 11. The WinRate is computed by assigning a weight of 1 to wins, 0.5 to ties, and 0 to losses. Formally:

$$\text{WinRate} = \frac{n_{\text{win}} + 0.5 \times n_{\text{tie}}}{n_{\text{win}} + n_{\text{tie}} + n_{\text{loss}}}, \tag{23}$$

where $n_{\text{win}}$, $n_{\text{tie}}$, and $n_{\text{loss}}$ denote the number of wins, ties, and losses, respectively. To mitigate position bias, we repeat the pairwise evaluation with swapped answer orders and report the averaged Win Rate. For the GPT-4 and RM baselines, high-quality preference pairs are selected based on GPT-4 score differences from UltraFeedback and reward differences computed by the OpenAssistant reward model[9], respectively.

We compare our LossDiff-IRM with several data-centric preference alignment methods, including:

- **CurriDPO (Pattnaik et al., 2024)** is a curriculum learning-based method, which orders preference pairs to organize curriculum learning based on various creteria, such as GPT4 score (CurriDPO-GPT4) and reward model score (CurriDPO-Reward Model).

---

[7]https://docs.vllm.ai/en/latest/

[8]https://bigmodel.cn/

[9]https://huggingface.co/OpenAssistant/oasst-rm-2.1-pythia-1.4b-epoch-2.5

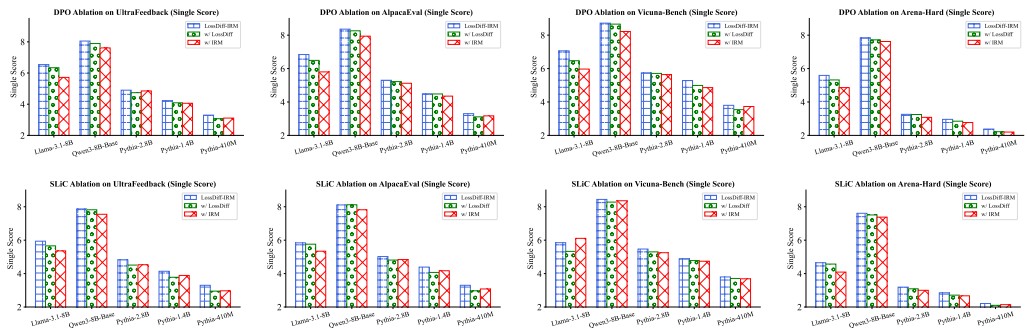

Figure 12: **Ablation study on Single Score.** Comparison of LossDiff-IRM and its ablated variants that select data relying solely on LossDiff ("w/ LossDiff") or solely on IRM ("w/ IRM"). *Top:* Training with DPO. *Bottom:* Training with SLiC. The ablation of WinRate is provided in Figure 3.

Table 10: **Concrete performance of ablation study of LossDiff-IRM with DPO**, which is corresponding to Figure 3 and Figure 12.

| LLM | DPO | Dataset Ratio | UltraFeedback | | AlpacaEval | | Vicuna-Bench | |
|---|---|---|---|---|---|---|---|---|
| | | | Single ↑ | WinRate ↑ | Single ↑ | WinRate ↑ | Single ↑ | WinRate ↑ |
| **Llama-3.1-8B** | LossDiff-IRM | 64% | **6.54** | **83.97** | **6.84** | **87.08** | **7.06** | **86.88** |
| | w/ LossDiff | 80% | 6.34 | 82.42 | 6.49 | 84.36 | 6.47 | 79.69 |
| | w/ IRM | 80% | 5.72 | 76.92 | 5.80 | 77.25 | 5.97 | 76.88 |
| **Qwen3-8B-Base** | LossDiff-IRM | 64% | **8.05** | **67.32** | **8.36** | **71.52** | **8.72** | **67.19** |
| | w/ LossDiff | 80% | 7.89 | 64.71 | 8.26 | 68.10 | 8.66 | 65.62 |
| | w/ IRM | 80% | 7.61 | 61.73 | 7.94 | 63.88 | 8.22 | 59.38 |
| **Pythia-2.8B** | LossDiff-IRM | 64% | **4.90** | **79.62** | **5.30** | **76.03** | **5.74** | **82.50** |
| | w/ LossDiff | 80% | 4.74 | 76.83 | 5.22 | 72.44 | 5.71 | 81.01 |
| | w/ IRM | 80% | 4.85 | 75.40 | 5.12 | 72.38 | 5.64 | 73.12 |
| **Pythia-1.4B** | LossDiff-IRM | 64% | **4.23** | **78.49** | **4.49** | **76.43** | **5.28** | **76.88** |
| | w/ LossDiff | 80% | 4.09 | 75.20 | 4.48 | 75.75 | 4.99 | 75.62 |
| | w/ IRM | 80% | 4.06 | 74.22 | 4.35 | 74.41 | 4.86 | 72.50 |
| **Pythia-410M** | LossDiff-IRM | 64% | **3.30** | **86.14** | **3.30** | **85.16** | **3.80** | **85.62** |
| | w/ LossDiff | 80% | 3.07 | 80.64 | 3.11 | 82.21 | 3.54 | 80.63 |
| | w/ IRM | 70% | 3.11 | 82.58 | 3.17 | 82.96 | 3.73 | 85.00 |

- $M_{\text{AP}}$ (**Huang et al., 2025**) is a margin-based preference data selection metric, called alignment potential, which integrates both explicit and implicit reward margins to quantify preference data.

- **RS-DPO** (**Khaki et al., 2024**) combines rejection sampling (RS) with DPO to generate high-quality preference data for DPO training. RS-DPO requires an external reward model to quantify preference pairs during rejection sampling.

# E ADDITIONAL EXPERIMENTAL RESULTS

## E.1 ABLATION STUDY: SINGLE SCORE

As a supplement to Figure 3, Figure 12 reports the Single Score for LossDiff-IRM and its two ablated variants: "w/ LossDiff" and "w/ IRM", which select data depending on LossDiff or IRM alone. Similar to the Win Rate results in Figure 3, the full LossDiff-IRM consistently outperforms its ablations. This results corroborates the higher Overlap Coefficients of LossDiff-IRM shown in Table 2 and Table 7, indicating that the combination of the two scoring functions provides a more reliable criterion for data selection than either one alone. Furthermore, Table 10 and Table 11 summarize the concrete performance of ablation studies with DPO and SLiC respectively.

Table 11: **Concrete performance of ablation studt of LossDiff-IRM with SLiC**, which is corresponding to Figure 3 and Figure 12.

| LLM | SLiC | Dataset Ratio | UltraFeedback | | AlpacaEval | | Vicuna-Bench | |
|---|---|---|---|---|---|---|---|---|
| | | | Single ↑ | WinRate ↑ | Single ↑ | WinRate ↑ | Single ↑ | WinRate ↑ |
| **Llama-3.1-8B** | LossDiff-IRM | 64% | **5.94** | **79.51** | **5.84** | **78.84** | **5.85** | **76.56** |
| | w/ LossDiff | 80% | 5.66 | 76.44 | 5.76 | 77.58 | 5.33 | 70.94 |
| | w/ IRM | 80% | 5.36 | 72.73 | 5.34 | 72.64 | 6.11 | 74.37 |
| **Qwen3-8B-Base** | LossDiff-IRM | 64% | **7.87** | **64.40** | **8.11** | **67.58** | **8.44** | **61.12** |
| | w/ LossDiff | 64% | 7.82 | 62.42 | 8.12 | 65.27 | 8.28 | 58.13 |
| | w/ IRM | 80% | 7.55 | 59.62 | 7.82 | 62.11 | 8.36 | 55.63 |
| **Pythia-2.8B** | LossDiff-IRM | 64% | **4.82** | **76.36** | **5.02** | **68.85** | 5.47 | **74.38** |
| | w/ LossDiff | 80% | 4.51 | 71.01 | 4.81 | 66.48 | 5.30 | 70.63 |
| | w/ IRM | 80% | 4.53 | 71.74 | 4.84 | 66.79 | 5.25 | 67.50 |
| **Pythia-1.4B** | LossDiff-IRM | 64% | **4.14** | **74.25** | **4.39** | **72.69** | **4.88** | **71.25** |
| | w/ LossDiff | 80% | 3.78 | 67.87 | 4.07 | 68.06 | 4.78 | 66.25 |
| | w/ IRM | 80% | 3.89 | 70.29 | 4.17 | 67.29 | 4.74 | 66.87 |
| **Pythia-410M** | LossDiff-IRM | 64% | **3.30** | **86.14** | **3.30** | **85.16** | **3.80** | **85.62** |
| | w/ LossDiff | 80% | 2.95 | 75.20 | 2.97 | 76.53 | 3.71 | 72.50 |
| | w/ IRM | 70% | 2.97 | 78.54 | 3.08 | 79.39 | 3.69 | 77.50 |

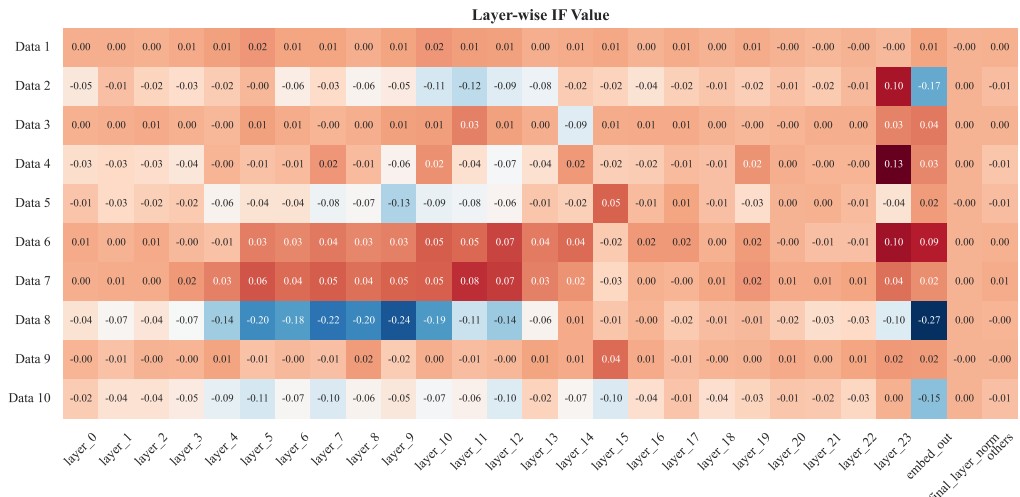

Figure 13: **Visualization of layer-wise IF value computed on Pythia-410M.**

## E.2 MORE VISUALIZATION OF IF

As a supplement to Figure 7, Figure 13 visualizes additional layer-wise IF values computed on the Pythia-410M model using the full UltraFeedback training and validation sets. Consistent with earlier observations, no single layer dominates the IF values, suggesting that approximating IF using gradients from only one or a few layers is not reliable and motivating the need for exploring effective correlated approximation proxies such as our LossDiff-IRM.

## E.3 RESULTS OF DIRECT TRAINING ON TRAIN, VAL, OR TRAIN+VAL SETS

Since this work introduces an additional validation set, a natural question is how models perform when trained directly on this validation set or on the union of training and validation sets. As a supplement, Table 12 reports the results of models trained on the training set, validation set, and the combined training + validation set, under both DPO and SLiC. We observe that none of these settings clearly outperforms the others, despite differences in dataset size. This highlights that existing datasets such as UltraFeedback contain a substantial amount of low-quality data, so randomly

Table 12: **Results of training on training, validation, or training + validation sets, respectively.**

| LLM | Training Data | Method | UltraFeedback | | AlpacaEval | | Vicuna-Bench | |
|---|---|---|---|---|---|---|---|---|
| | | | Single ↑ | WinRate ↑ | Single ↑ | WinRate ↑ | Single ↑ | WinRate ↑ |
| **Llama-3.1-8B** | Val | DPO | 5.28 | 72.77 | 5.34 | 73.26 | 5.83 | 73.44 |
| | Train + Val | DPO | 5.50 | 75.16 | 5.47 | 73.24 | 5.50 | 71.95 |
| | Train | DPO | 5.77 | 77.61 | 5.87 | 78.41 | 6.04 | 73.75 |
| | Train | DPO + LossDiff-IRM | **6.54** | **83.97** | **6.84** | **87.08** | **7.06** | **86.88** |
| | Val | SLiC | 5.19 | 72.50 | 5.16 | 72.39 | 5.29 | 67.50 |
| | Train + Val | SLiC | 5.03 | 67.41 | 5.07 | 69.82 | 5.15 | 66.56 |
| | Train | SLiC | 5.09 | 70.72 | 5.13 | 72.13 | 5.40 | 71.88 |
| | Train | SLiC + LossDiff-IRM | **5.94** | **79.51** | **5.84** | **78.84** | **5.85** | **76.56** |
| **Qwen3-8B-Base** | Val | DPO | 7.83 | 61.73 | 8.07 | 67.81 | 8.36 | 65.33 |
| | Train + Val | DPO | 7.72 | 62.01 | 7.88 | 63.77 | 8.18 | 52.50 |
| | Train | DPO | 7.64 | 61.41 | 7.92 | 63.85 | 8.21 | 62.14 |
| | Train | DPO + LossDiff-IRM | **8.05** | **67.32** | **8.36** | **71.52** | **8.72** | **67.19** |
| | Val | SLiC | 7.71 | 60.62 | 8.07 | 64.59 | 8.38 | 58.44 |
| | Train + Val | SLiC | 7.56 | 58.44 | 7.78 | 60.93 | 8.22 | 60.59 |
| | Train | SLiC | 7.55 | 59.54 | 7.61 | 59.71 | 8.05 | 54.69 |
| | Train | SLiC + LossDiff-IRM | **7.87** | **64.40** | **8.11** | **67.58** | **8.44** | **61.12** |
| **Pythia-2.8B** | Val | DPO | 4.63 | 70.88 | 4.80 | 64.51 | 5.49 | 69.37 |
| | Train + Val | DPO | 4.54 | 71.39 | 4.86 | 63.67 | 5.35 | 66.25 |
| | Train | DPO | 4.60 | 70.53 | 4.95 | 67.50 | 5.35 | 67.50 |
| | Train | DPO + LossDiff-IRM | **4.90** | **79.62** | **5.30** | **76.03** | **5.74** | **82.50** |
| | Val | SLiC | 4.42 | 67.70 | 4.71 | 62.11 | 4.89 | 61.88 |
| | Train + Val | SLiC | 4.33 | 64.29 | 4.63 | 57.49 | 4.84 | 56.25 |
| | Train | SLiC | 4.36 | 67.46 | 4.48 | 61.66 | 4.90 | 65.00 |
| | Train | SLiC + LossDiff-IRM | **4.82** | **76.36** | **5.02** | **68.85** | **5.47** | **74.38** |
| **Pythia-1.4B** | Val | DPO | 3.69 | 66.45 | 4.06 | 66.22 | 4.65 | 65.65 |
| | Train + Val | DPO | 3.80 | 65.50 | 4.01 | 64.30 | 4.69 | 65.62 |
| | Train | DPO | 3.70 | 65.88 | 3.99 | 66.17 | 4.71 | 64.38 |
| | Train | DPO + LossDiff-IRM | **4.23** | **78.49** | **4.49** | **76.43** | **5.28** | **76.88** |
| | Val | SLiC | 3.76 | 66.75 | 4.02 | 63.33 | 4.64 | 59.38 |
| | Train + Val | SLiC | 3.68 | 63.67 | 3.98 | 63.12 | 4.45 | 59.38 |
| | Train | SLiC | 3.66 | 63.58 | 3.98 | 63.68 | 4.67 | 60.00 |
| | Train | SLiC + LossDiff-IRM | **4.14** | **74.25** | **4.39** | **72.69** | **4.88** | **71.25** |
| **Pythia-410M** | Val | DPO | 2.72 | 72.57 | 2.77 | 73.04 | 3.74 | 72.50 |
| | Train + Val | DPO | 2.77 | 73.62 | 2.80 | 73.85 | 3.31 | 73.75 |
| | Train | DPO | 2.81 | 75.25 | 2.77 | 73.51 | 3.10 | 69.37 |
| | Train | DPO + LossDiff-IRM | **3.30** | **86.14** | **3.30** | **85.16** | **3.80** | **85.62** |
| | Val | SLiC | 2.68 | 71.13 | 2.73 | 71.74 | 3.51 | 66.25 |
| | Train + Val | SLiC | 2.83 | 73.17 | 2.84 | 73.60 | 3.21 | 63.12 |
| | Train | SLiC | 2.81 | 73.80 | 2.82 | 73.91 | 3.31 | 71.25 |
| | Train | SLiC + LossDiff-IRM | **3.07** | **80.08** | **3.09** | **84.39** | **3.83** | **79.36** |

Table 13: **Performance of introducing more validation data.** The validation set is extended by introducing validation set from OASST (Köpf et al., 2023) and GoldenHH (Cai et al., 2023).

| LLM | Validation Dataset | DPO | UltraFeedback | | AlpacaEval | | Vicuna-Bench | |
|---|---|---|---|---|---|---|---|---|
| | | | Single ↑ | WinRate ↑ | Single ↑ | WinRate ↑ | Single ↑ | WinRate ↑ |
| **Pythia-2.8B** | N/A | Full Data | 4.60 | 70.53 | 4.95 | 67.50 | 5.35 | 67.50 |
| | Ultra | LossDiff-IRM | 4.90 | 79.62 | **5.30** | **76.03** | 5.74 | **82.50** |
| | Ultra + OASST | LossDiff-IRM | 4.88 | 78.71 | 5.11 | 74.16 | 5.85 | 76.88 |
| | Ultra + GoldenHH | LossDiff-IRM | **4.99** | **80.07** | 5.28 | 74.06 | **5.89** | 74.38 |
| **Pythia-1.4B** | N/A | Full Data | 3.70 | 65.88 | 3.99 | 66.17 | 4.71 | 64.38 |
| | Ultra | LossDiff-IRM | 4.23 | 78.49 | 4.49 | 76.43 | 5.28 | 76.88 |
| | Ultra + OASST | LossDiff-IRM | 4.17 | 77.26 | 4.44 | 77.11 | 4.91 | 77.22 |
| | Ultra + GoldenHH | LossDiff-IRM | **4.36** | **80.12** | **4.69** | **79.02** | **5.39** | **81.01** |
| **Pythia-410M** | N/A | Full Data | 2.81 | 75.25 | 2.77 | 73.51 | 3.10 | 69.37 |
| | Ultra | LossDiff-IRM | **3.30** | **86.14** | **3.30** | 85.16 | 3.80 | 85.62 |
| | Ultra + OASST | LossDiff-IRM | 3.11 | 84.77 | 3.19 | 85.24 | **3.96** | **86.25** |
| | Ultra + GoldenHH | LossDiff-IRM | 3.25 | 84.62 | 3.28 | **86.71** | 3.73 | 83.75 |

splitting out a validation set or simply enlarging the training set by taking their union does not yield performance gains. In contrast, when applying our LossDiff-IRM data selection on the training set, performance improves significantly, further confirming that LossDiff-IRM effectively identifies the valuable preference data beneficial for current model alignment.

Table 14: **Time cost analysis between RS-DPO and LossDiff-IRM.**

| Trained Model | # Prompts | RS-DPO | | LossDiff-IRM | |
|---|---|---|---|---|---|
| | | Time | Throughput (prompts/sec) | Time | Throughput (prompts/sec) |
| Llama-3.1-8B | 48,908 | 4 h 34 min | 2.97 | 2 h 14 min | 6.08 |
| Qwen3-8B-Base | 48,908 | 4 h 50 min | 2.81 | 2h 43 min | 5.03 |

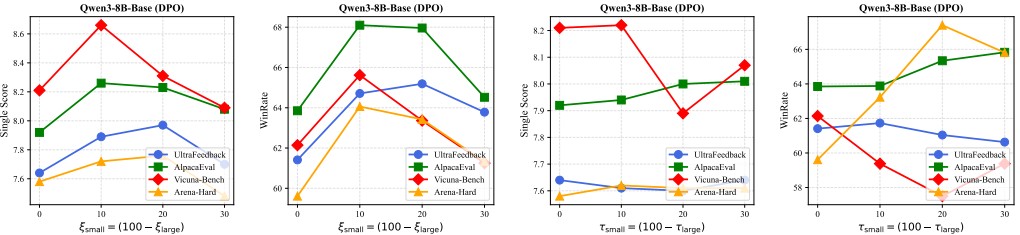

Figure 14: **Analysis of percentile thresholds $\xi_{small}, \xi_{large}, \tau_{small}, \tau_{large}$ of Qwen3-8B-Base.** We vary $\xi_{small} = (100 - \xi_{large})$ and $\tau_{small} = (100 - \tau_{large})$ with in $\{0, 10, 20, 30\}$. Analysis for Llama-3.1-8B is illustrated in Figure 6.

### E.4 IMPACT OF INTRODUCING MORE VALIDATION DATA

Since LossDiff-IRM relies on a validation set as a reference for data selection, Figure 4 has examined the effect of noisy validation sets. A remaining question is whether enlarging the validation set by incorporating data from multiple sources can further improve its effectiveness. As a supplement, Table 13 reports the performance of LossDiff-IRM when the original Ultra validation set is extended with additional data from OASST (Köpf et al., 2023) and GoldenHH (Cai et al., 2023), resulting in {Ultra + OASST} and {Ultra + GoldenHH}, each containing 22,227 preference pairs. We observe that extending Ultra with OASST or GoldenHH yields only marginal improvements on certain models or benchmarks, without consistent or significant gains. Notably, regardless of whether the validation set is Ultra, {Ultra + OASST}, or {Ultra + GoldenHH}, applying LossDiff-IRM consistently outperforms training on the full dataset. These results suggest that a large validation preference set is not strictly necessary for effective LossDiff-IRM selection. LossDiff-IRM remains certain robustness even when the validation set differs from the training distribution, as long as it provides a rough reference direction for distinguishing preference data quality.

### E.5 TIME COST ANALYSIS BETWEEN LOSSDIFF-IRM AND RS-DPO

High-quality preference data selection and generation are two types of data-centric methods. For example, RS-DPO (Khaki et al., 2024) focuses on generating new high-quality preference pairs through rejection sampling using the SFT model, whereas our LossDiff-IRM focuses on analyzing existing annotated preference pairs to identify valuable preference pairs. Specifically, RS-DPO requires an external reward model to evaluate the responses generated by the SFT model. Given $N$ prompts and $K$ sampled responses per prompt, RS-DPO needs to score $N \times K$ responses with the reward model in order to construct contrastive preference pairs. For LossDiff-IRM, it requires scoring all preference pairs with both the current model and an auxiliary model. For the same scale of $N$ preference pairs, LossDiff-IRM requires $N$ forward passes with the current model and $N$ forward passes with the auxiliary model, resulting in a total of $2N$ forward passes, where the two models share the same architecture and differ only in parameters. To facilitate a direct comparison, Table 14 reports the data processing time cost of RS-DPO (including data generation and rejection sampling) using the OpenAssistant-1.4B reward model with K=8, as well as the data-selection time of LossDiff-IRM. All experiments were conducted on a single H100-80GB GPU.

The dominant cost of RS-DPO comes from using the reward model to evaluate all $N \times K$ responses generated by the SFT model. Even using a relatively small 1.4B reward model, the time cost is actually higher than LossDiff-IRM. Therefore, the time cost of LossDiff-IRM is at least on par with RS-DPO, and even often smaller, especially when considering that RS-DPO scales with $K$ or uses a larger reward model.

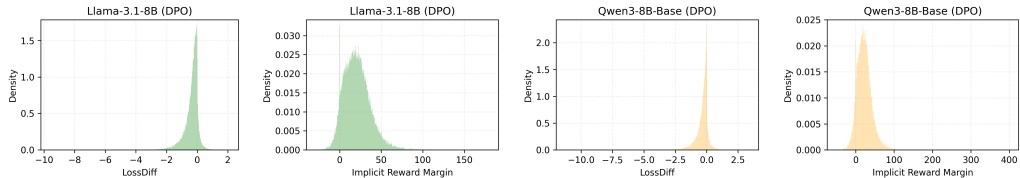

Figure 15: **Visualization of distribution of LossDiff and implicit reward margin** for Llama-3.1-8B and Qwen3-8B-Base, respectively.

Table 15: **Concrete performance of LossDiff-IRM with noisy validation set**, which is corresponding to Figure 4.

| Method | UltraFeedback | | AlpacaEval | | Vicuna-Bench | |
|---|---|---|---|---|---|---|
| | Single ↑ | WinRate ↑ | Single ↑ | WinRate ↑ | Single ↑ | WinRate ↑ |
| *Llama-3.1-8B* | | | | | | |
| Full Data | 5.77 | 77.61 | 5.87 | 78.41 | 6.04 | 73.75 |
| LossDiff-IRM | **6.54** | **83.97** | **6.84** | **87.08** | **7.06** | **86.88** |
| + Noise Rate = 0.1 | 6.44 | 83.79 | 6.69 | 85.99 | 6.76 | 81.56 |
| + Noise Rate = 0.2 | 6.19 | 80.76 | 6.35 | 81.30 | 6.67 | 78.44 |
| + Noise Rate = 0.3 | 6.19 | 80.71 | 6.24 | 82.03 | 6.54 | 82.50 |
| + Noise Rate = 0.4 | 5.95 | 79.02 | 6.01 | 80.37 | 6.54 | 78.75 |
| *Qwen3-8B-Base* | | | | | | |
| Full Data | 7.64 | 61.41 | 7.92 | 63.85 | 8.21 | 62.14 |
| LossDiff-IRM | **8.05** | **67.32** | **8.36** | **71.52** | **8.72** | **67.19** |
| + Noise Rate = 0.1 | 7.70 | 63.78 | 8.11 | 68.75 | 8.47 | 65.62 |
| + Noise Rate = 0.2 | 7.99 | 66.70 | 8.26 | 69.57 | 8.61 | 61.25 |
| + Noise Rate = 0.3 | 7.97 | 65.98 | 8.15 | 69.16 | 8.65 | 66.25 |
| + Noise Rate = 0.4 | 7.84 | 63.90 | 8.26 | 68.35 | 8.49 | 65.62 |

### E.6 MORE ANALYSIS OF PERCENTILE THRESHOLDS

As a supplement, Figure 14 provides the analysis of percentile thresholds $\xi_{small}, \xi_{large}, \tau_{small}, \tau_{large}$ for Qwen3-8B-Base. The rough trends are similar as for Llama-3.1-8B: performance first improves and then degrades as the thresholds become stricter. Following this observation, we finally set $\xi_{small}$ and $\xi_{small}$ as 10, $\tau_{small}$ and $\tau_{small}$ as 10 for all models.

### E.7 VISUALIZATION OF DISTRIBUTION OF LOSSDIFF AND IRM

Figure 15 visualizes the LossDiff and IRM distributions for Llama-3.1-8B and Qwen3-8B-Base, respectively. We observe that both LossDiff and IRM exhibit distributional shapes resembling unimodal, Gaussina-like distributions, where the majority of data lies in the middle region and only a small number of data fall in the extremes. In our data selection strategy, both criteria concentrate data in their middle ranges, taking their intersection would not result in an extremely small subset. The overlap between their medium regions remains sufficiently large and high-quality for alignment.

### E.8 ADDITIONAL SUPPLEMENT

As a supplement to Figure 3 and Figure 12, we report the corresponding numerical results in Table 10. In addition, Figure 4 shows the impact of noisy validation sets on LossDiff-IRM under different noise rates, with the detailed results provided in Table 15.

