# OpenReview forum: "Towards Understanding Valuable Preference Data for Large Language Model Alignment"
_ICLR.cc/2026/Conference — ICLR 2026 Poster_

### Official Review · Reviewer_sQAd · 2025-10-25

**Soundness:** 3
**Presentation:** 3
**Contribution:** 3
**Rating:** 6
**Confidence:** 4

**Summary:**

This paper studies the data quality of preference pairs for LLM alignment.
The authors begin with influence function analysis of each preference pair's impact, which measures the consistency of datapoint's gradient with the overall gradient of validation set.
The analysis shows the medium-IF preference pairs are more valuable, so the authors design the TIF metric: $\mathbb I[\delta_{low} < IF < \delta_{high}]$ to select medium-IF data for training.
Furthermore, to reduce the computation cost of IF, the authors introduce two simplified data quality metrics:
1. LossDiff: the difference of loss between $\pi_\theta$ and a validation model $\pi_{val}$;
2. IRM: the implict reward margin.

Both metrics correlate well with IF and can be used to approximate TIF for data selection.
Empirical results on different models, DPO variants, and benchmarks demonstrate the effectiveness of the proposed data selection methods in improving alignment performance.

**Strengths:**

1. **Clear Motivation and Well-Justified Methods**: Data quality metrics based on model state serve as an intuitive and reasonable approach to improve preference optimization performance. The paper's derivation from influence function analysis to practical data selection metrics is well-motivated and clearly presented.
2. **Stable Improvements**: The proposed data selection methods consistently enhance alignment performance across different models, DPO variants, and benchmarks, demonstrating their robustness and effectiveness.
3. **Comprehensive Ablation**: The paper includes several ablation studies to provide further insights including model-specific selection, noisy robustness, and optimized layers.
4. **Presentation Quality**: The paper is well-written and easy to follow, with clear explanations of the proposed methods and experimental results.

**Weaknesses:**

My main concern with this work is the lack of comparison with prior model-specific data selection methods.

The authors state in L56-58:
> Our above analysis suggests a reasonable yet seldom-discussed viewpoint: preference data selection
should be performed for specific models and explicitly related to the training process

However, there are already several works discussing model-specific data selection. For example, implicit margin based selection methods [1,2,3,4] also take the model state $\pi_\theta$ into consideration when selecting data.
So I think it would be better not to claim that this is a "seldom-discussed viewpoint".

Moreover, [3,4] utilize IRM to select data in a similar way: They prioritize preference pairs with the smallest absolute IRM value, i.e., medium-IRM data, which is similar to the TIF metric proposed in this paper.
So it would be better to cite and compare with these prior works.

---

**References**

[1] Morimura, Tetsuro, et al. "Filtered Direct Preference Optimization." Proceedings of the 2024 Conference on Empirical Methods in Natural Language Processing. 2024.\
[2] Deng, Xun, et al. "Less is more: Improving llm alignment via preference data selection." arXiv preprint arXiv:2502.14560 (2025).\
[3] Huang, Kexin, et al. "Larger or Smaller Reward Margins to Select Preferences for LLM Alignment?." Forty-second International Conference on Machine Learning. \
[4] Yang, Sen, et al. "Not All Preference Pairs Are Created Equal: A Recipe for Annotation-Efficient Iterative Preference Learning." Findings of the Association for Computational Linguistics: EMNLP 2024. 2024.

**Questions:**

Can you provide more comparison with prior model-specific data selection methods?

---

> ### Author Response · Authors · 2025-11-25
> **Response to reviewer sQAd (Part 1/2)**
>
> We sincerely thank the reviewer for the time and effort spent reviewing our paper and for acknowledging the clear motivation and well-justified method, stable improvement, comprehensive ablation, and presentation quality. Below, we address the raised concerns and questions in detail:
>
> > **W1:** My main concern with this work is the lack of comparison with prior model-specific data selection methods.
> The authors state in L56-58:
> Our above analysis suggests a reasonable yet seldom-discussed viewpoint: preference data selection should be performed for specific models and explicitly related to the training process
> However, there are already several works discussing model-specific data selection. For example, implicit margin based selection methods [1,2,3,4] also take the model state  into consideration when selecting data. So I think it would be better not to claim that this is a "seldom-discussed viewpoint".
> Moreover, [3,4] utilize IRM to select data in a similar way: They prioritize preference pairs with the smallest absolute IRM value, i.e., medium-IRM data, which is similar to the TIF metric proposed in this paper. So it would be better to cite and compare with these prior works.
> [1] Morimura, Tetsuro, et al. "Filtered Direct Preference Optimization." Proceedings of the 2024 Conference on Empirical Methods in Natural Language Processing. 2024.
> [2] Deng, Xun, et al. "Less is more: Improving llm alignment via preference data selection." arXiv preprint arXiv:2502.14560 (2025).
> [3] Huang, Kexin, et al. "Larger or Smaller Reward Margins to Select Preferences for LLM Alignment?." Forty-second International Conference on Machine Learning.
> [4] Yang, Sen, et al. "Not All Preference Pairs Are Created Equal: A Recipe for Annotation-Efficient Iterative Preference Learning." Findings of the Association for Computational Linguistics: EMNLP 2024. 2024.
> **Q1:** Can you provide more comparison with prior model-specific data selection methods?
>
> **A:** Thanks for your valuable comments and question. We carefully read these papers provided by the reviewer, and **we appreciate the reviewer that these studies are very relevant to our research work. We have cited them in our revised manuscript**. Specifically, Morimura et al. [1] propose the fDPO that uses a pretrained reward model to monitor the quality of preference data during DPO training. Deng et al. [2] propose a magin-based principle that adopts both external margin from reward model and internal margin from DPO implicit reward for data selection. Huang et al. [3] propose the alignment potential metric $M_\text{AP}$, which combines the explict reward margin with current implict reward margin to quantify the quality of preference data. Yang et al. [4] adopt the implicit reward margin of DPO as measure to selection small-margin data for annotation-efficient preference training.
>
> Overall, these approaches mainly center on **reward margin-based criteria** for data selection. Some methods rely solely on implicit reward margin [4] or external reward model [1], while some methods consider both of them [2,3]. Although they discuss the implicit reward margin that is a model-specific assessment for preference data selection, **their focus and analysis are primarily on deciding which reward margins are beneficial for training**.
>
> Notably, our starting point and analysis are different from them. **We employ influence function (IF) as a principled analytical tool to study preference data from the viewpoint of generalization rather than reward margin magnitude**. Through IF-based analysis, we demonstrate a new insight: **medium-IF preference pairs contribute more effectively to preference generalization of alignment training**. The implicit reward margin is one of the scoring functions we use to approximate IF in practice. Our analysis also identifies loss difference as another effective IF approximation, and potentially other metrics with strong positive correlation to IF may also serve as effective scoring functions. Therefore, in addition to the data selection strategy, our core contribution lies in providing an analytical pipeline that connects a classical machine-learning tool IF with the generalization properties of preference alignment.

---

> ### Author Response · Authors · 2025-11-25
> **Response to reviewer sQAd (Part 2/2)**
>
> [continued to **W1** and **Q1**]
>
> Following the reviewer's suggestion, **we supplement the baselines including CurriDPO [5], $M_\text{AP}$ [3] and RS-DPO [4] for comparisons**. CurriDPO orders preference pairs to organize curriculum learning based on creteria such as GPT4 score and reward model score. Beyond the three benchmarks used previously (UltraFeedback, AlpacaEval, and Vicuna-Bench), we further add the Arena-Hard benchmark for more comprehensive evaluation. The results are summarized as follows:
>
> |Llama-3.1-8B|UltraFeedback Single|WinRate|AlpacaEval Single|WinRate|Vicuna-Bench Single|WinRate|Arena-Hard Single|WinRate|
> |-|-|-|-|-|-|-|-|-|
> |CurriDPO-GPT4|5.47|74.23|5.53|75.01|5.84|74.06|4.55|79.77|
> |CurriDPO-Reward Model|5.51|74.62|5.54|74.29|5.59|74.06|4.62|79.49|
> |$M_\text{AP}$|4.92|67.71|4.84|67.54|4.61|59.06|3.92|69.94|
> |RS-DPO|5.70|75.98|6.39|84.84|7.04|82.81|4.69|82.05|
> |**LossDiff-IRM (Ours)**|**6.54**|**83.97**|**6.84**|**87.08**|**7.06**|**86.88**|**5.59**|**88.40**|
>
> |Qwen3-8B-Base|UltraFeedback Single|WinRate|AlpacaEval Single|WinRate|Vicuna-Bench Single|WinRate|Arena-Hard Single|WinRate|
> |-|-|-|-|-|-|-|-|-|
> |CurriDPO-GPT4|7.61|61.04|7.74|62.35|8.16|54.69|7.52|62.51|
> |CurriDPO-Reward Model|7.60|59.62|7.84|61.96|8.20|61.58|7.55|63.97|
> |$M_\text{AP}$|7.48|58.81|7.76|60.91|8.09|52.27|7.44|59.78|
> |RS-DPO|7.88|64.87|**8.36**|**74.07**|**8.93**|**71.90**|7.48|61.32|
> |**LossDiff-IRM (Ours)**|**8.05**|**67.32**|**8.36**|71.52|8.72|67.19|**7.83**|**68.63**|
>
> From the results, it can be observed that **our LossDiff-IRM outperforms these competitors on the most cases**. Specifically, the average improvement is +2.62% on WinRate over the best baselines across two models, especially achieving improvements of +6.45% on average on Llama-3.8-8B. **These results suggest the effectiveness of our influence function (IF)-driven analysis and LossDiff-IRM data selection method**. Additionally, we observe that RS-DPO, which generates model-specific preference pairs via rejection sampling using the SFT model, serves as a strong competitor. This finding aligns with our idea that the model's perspective is an important factor in determining whether a preference pair is valuable. We will include these results in the revised manuscript.
>
> [1] Morimura T, Sakamoto M, Jinnai Y, et al. Filtered Direct Preference Optimization[C]//Proceedings of the 2024 Conference on Empirical Methods in Natural Language Processing. Association for Computational Linguistics, 2024: 22729-22770.
> [2] Deng X, Zhong H, Ai R, et al. Less is more: Improving llm alignment via preference data selection[J]. arXiv preprint arXiv:2502.14560, 2025.
> [3] Huang K, Wu J, Chen Z, et al. Larger or Smaller Reward Margins to Select Preferences for LLM Alignment?[C]//Forty-second International Conference on Machine Learning.
> [4] Yang S, Cui L, Cai D, et al. Not All Preference Pairs Are Created Equal: A Recipe for Annotation-Efficient Iterative Preference Learning[C]//Findings of the Association for Computational Linguistics: EMNLP 2024. 2024: 6549-6561.
> [5] Pattnaik P, Maheshwary R, Ogueji K, et al. Enhancing alignment using curriculum learning & ranked preferences[C]//Findings of the Association for Computational Linguistics: EMNLP 2024. 2024: 12891-12907.

---

> ### Comment · Reviewer_sQAd · 2025-11-25
>
> Thanks for your detailed responses and the additional experiments.
>
> As I've mentioned, *"[3,4] utilize IRM to select data in a similar way: They prioritize preference pairs with the smallest absolute IRM value, i.e., medium-IRM data"*, so I believe they would perform at least similarly to your "w/ LossDiff" baseline.
> But it seems that [3] performs worse than "w/ LossDiff" in you rebuttal experiments.
> Could you please explain the possible reasons for this discrepancy?
>
> Moreover, the method in [4] can also be utilized for data selection, and is very simple to implement, which only replaces your "w/ LossDiff" baseline's selection criterion: $\mathbb I[\tau_1<IRM < \tau_2]$
> with the absolute IRM value:
> $\mathbb I[|IRM| < \tau]$.
> And [3] augments [4] with an additional reward margin score, with a tunable hyperparameter to balance the two margins.
>
> Therefore, a most direct comparison would be to include the absolute IRM-based selection as a baseline, which is simple to implement and has been shown to be effective [3,4].
> Could you please include this baseline in your experiments and report the results?
>
> BTW, in your rebuttal you mentioned:
> > *Following the reviewer's suggestion, we supplement the baselines including CurriDPO [5], [3] and RS-DPO [4] for comparisons.*
>
> There seems to be a typo here, as RS-DPO is not proposed in [4] *(Yang S, Cui L, Cai D, et al. Not All Preference Pairs Are Created Equal: A Recipe for Annotation-Efficient Iterative Preference Learning)*.

---

> > ### Author Response · Authors · 2025-11-26
> > **Response to reviewer sQAd**
> >
> > Thanks for your constructive feedback regarding the baselines. The alignment potential ($M_\text{AP}$) proposed in [3] is defined as follows:
> > $$M_\text{AP}=|r_\text{explicit}(x,y_w)-r_\text{explicit}(x,y_l)|-|r_\text{implicit}^{(\theta)}(x,y_w)-r_\text{implicit}^{(\theta)}(x,y_l)|,$$
> > where the first term denotes the explicit reward margin provided by a reward model, and the second term denotes the implicit reward margin (IRM). In [3], **preference pairs with higher $M_\text{AP}$ values** are regarded as higher-quality data. As the reviewer correctly pointed out, the second term favors preference pairs with **smaller absolute IRM** (i.e., medium-IRM data). However, $M_\text{AP}$ is simultaneously influenced by the first term, which prioritizes preference pairs with **larger explicit reward margins**, making the method dependent on the chosen external reward model.
> >
> > In our initial implementation, we used the OpenAssistant-1.4B reward model. After the reviewer’s helpful reminder, **we immediately re-implement $M_\text{AP}$ using the same reward model "ArmoRM-Llama3-8B-v0.1" same as in [3] and re-run the experiments**. In addtion, following the reviewer's suggestion, we supplement the competitor Abs-IRM that selects preference pairs based on absolute IRM value, which can be viewed as an ablation of $M_\text{AP}$ that removes the explicit reward margin term. The results are reported as follows:
> >
> > |Llama-3.1-8B|UltraFeedback Single|WinRate|AlpacaEval Single|WinRate|Vicuna-Bench Single|WinRate|Arena-Hard Single|WinRate|
> > |-|-|-|-|-|-|-|-|-|
> > |Abs-IRM|5.86|78.16|5.86|77.48|6.42|77.96|4.66|80.50|
> > |$M_\text{AP}$|6.04|79.99|6.21|80.88|6.34|74.69|5.08|85.30|
> > |**LossDiff-IRM (Ours)**|**6.54**|**83.97**|**6.84**|**87.08**|**7.06**|**86.88**|**5.59**|**88.40**|
> >
> > |Qwen3-8B-Base|UltraFeedback Single|WinRate|AlpacaEval Single|WinRate|Vicuna-Bench Single|WinRate|Arena-Hard Single|WinRate|
> > |-|-|-|-|-|-|-|-|-|
> > |Abs-IRM|7.58|60.88|8.00|65.08|7.89|56.25|7.61|67.12|
> > |$M_\text{AP}$|7.95|**67.84**|8.31|71.11|8.62|66.87|7.72|66.55|
> > |**LossDiff-IRM (Ours)**|**8.05**|67.32|**8.36**|**71.52**|**8.72**|**67.19**|**7.83**|**68.63**|
> >
> > From the above results, we observe that the **$M_\text{AP}$ is also a strong competitor**, with performance substantially improved after switching to the 8B-level reward model. MAP also outperforms Abs-IRM, as Abs-IRM can be viewed as an ablation of $M_\text{AP}$ that removes the additional explicit reward margin. This suggests that reward model choice is important for $M_\text{AP}$ due to its reliance on explicit reward margins. Additionally, LossDiff-IRM without relying on external reward model, is better than $M_\text{AP}$ on most cases, suggesting **the effectiveness of our influence function (IF)–motivated idea and the LossDiff-IRM data selection strategy**. We will include these results in the revised manuscript.
> >
> > Finally, we thank the reviewer very much for pointing out the typo. RS-DPO is proposed in [6], “RS-DPO: A Hybrid Rejection Sampling and Direct Preference Optimization Method for Alignment of Large Language Models”. We have corrected this error in the response.
> >
> > If you have any concerns or further suggestions, we would be glad to engage in further discussion.
> >
> > [6] Khaki S, Li J J, Ma L, et al. RS-DPO: A Hybrid Rejection Sampling and Direct Preference Optimization Method for Alignment of Large Language Models[C]//Findings of the Association for Computational Linguistics: NAACL 2024. 2024: 1665-1680.

---

> ### Comment · Reviewer_sQAd · 2025-11-27
>
> Thanks for your updated results and clarifications.
> I still have one more question regarding the comparison with absolute IRM-based selection:
> The concepts of medium-IRM selection and Abs-IRM selection are very similar, which you also agree by pointing out: *"smaller absolute IRM (i.e., medium-IRM data)"*.
> However, in your updated results, Abs-IRM seems to perform clearly worse than your w/LossDiff baseline (Figure 3). For example, 77.48 vs. around 85 for Llama 3.1 on Alpaca winrate.
>
> There might be two possible reasons for this discrepancy:
> 1. Is it because of different selection ratios for Abs-IRM and w/LossDiff in the updated experiments? If so, a fair comparison with the same selection ratio is needed.
> 2. Or the mean of IRM is not close to zero in your experiments, making Abs-IRM not equivalent to medium-IRM selection? If this is the case, will re-centering the IRM values before applying Abs-IRM selection improve its performance?
>
> If none of the above two reasons apply, could you please provide more analysis on why the two similar methods perform differently? For example, you can show the overlap rate of selected data between the two methods and analyze the differences.
>
> While the concept of medium-IRM in your work is similar to Abs-IRM in previous study, the introduction of IRM here is based on comprehensive influence function analysis, which provides a more solid foundation for data selection.
> So I still recognize the value of your proposed method, but clarifying the above discrepancy will further improve the rigor of your experimental comparisons.

---

> ### Author Response · Authors · 2025-11-28
> **Response to reviewer sQAd**
>
> Thanks for your constructive feedback regarding the baselines.
>
> Firstly, **we would like to clarify that "w/" means "with". Our ablation "w/ LossDiff" uses only loss difference (LossDiff) for preference data selection, and the other ablation "w/ IRM" uses only implicit reward margin (IRM) to select preference data**. Therefore, Abs-IRM, which selects preference pairs with smaller absolute IRM values, is similar to “w/ IRM”, which selects medium-IRM data.
>
> Regarding the selection ratios, the reviewer’s understanding is correct. “w/ LossDiff” and “w/ IRM” use about 80% of data, whereas Abs-IRM uses about 64% that is same as final LossDiff-IRM setting for fair comparison. Following the reviewer's suggestion, **we immediately re-run an additional Abs-IRM with 80% data, matching our two ablations**. As for the mean issue, it is also correct that the raw IRM distribution is not centered around zero; in our implementation, we have re-centered IRM before selection, so this is not the source of the discrepancy.
>
> Following the reviewer's suggestion, **we now provide the results of Abs-IRM with an 80% ratio, together with the concrete results of “w/ LossDiff” and “w/ IRM” for a clear comparison**. The results are summarized below:
>
> |Llama-3.1-8B|Data Ratio|UltraFeedback Single|WinRate|AlpacaEval Single|WinRate|Vicuna-Bench Single|WinRate|Arena-Hard Single|WinRate|
> |-|-|-|-|-|-|-|-|-|-|
> |Abs-IRM|64%|5.86|78.16|5.86|77.48|6.42|77.96|4.66|80.50|
> |Abs-IRM|80%|5.72|77.09|5.77|77.42|5.92|70.94|4.63|80.62|
> |w/ IRM|80%|5.72|76.92|5.80|77.25|5.97|76.88|4.86|81.55|
> |w/ LossDiff|80%|6.34|82.42|6.49|84.36|6.47|79.69|5.32|87.29|
> |LossDiff-IRM|64%|**6.54**|**83.97**|**6.84**|**87.08**|**7.06**|**86.88**|**5.59**|**88.40**|
>
> |Qwen3-8B-Base|Data Ratio|UltraFeedback Single|WinRate|AlpacaEval Single|WinRate|Vicuna-Bench Single|WinRate|Arena-Hard Single|WinRate|
> |-|-|-|-|-|-|-|-|-|-|
> |Abs-IRM|64%|7.58|60.88|8.00|65.08|7.89|56.25|7.61|67.12|
> |Abs-IRM|80%|7.74|63.16|8.02|66.37|8.11|55.63|7.66|64.73|
> |w/ IRM|80%|7.61|61.73|7.94|63.88|8.22|59.38|7.62|63.23|
> |w/ LossDiff|80%|7.89|64.71|8.26|68.10|8.66|65.62|7.72|64.06|
> |LossDiff-IRM|64%|**8.05**|**67.32**|**8.36**|**71.52**|**8.72**|**67.19**|**7.83**|**68.63**|
>
> The reviewer's undertanding is correct that **“smaller absolute IRM” and “medium IRM” follow similar principles, so Abs-IRM and “w/ IRM” show comparable performance**; "w/ LossDiff" performs relatively better performance over "Abs-IRM" and "w/ IRM", because LossDiff leverages an additional validation set to guide data selection, which Abs-IRM and “w/ IRM” do not use. Moreover, **LossDiff-IRM, which combines both LossDiff and IRM, further outperforms its two ablations**, supporting our idea that combining the two scoring functions helps offset the specific errors each incurs when approximating the exact influence function (IF) for data selection.
>
> Thanks again for your valuable advice and support to help us improve the quality of our manuscript. If you have futher questions or concerns, we wil be glad to engage futher discussions with you.

---

### Official Review · Reviewer_CD9n · 2025-10-29

**Soundness:** 3
**Presentation:** 3
**Contribution:** 3
**Rating:** 6
**Confidence:** 4

**Summary:**

This paper proposes a novel data selection method for preference data, inspired by two points:

1. Preference data selection based on influence function (IF);

2. DPO formulation implicitly encodes the reward difference, which can be leveraged for better estimation of IF.

Through preliminary experiments, the authors reveal an interesting phenomenon: data with medium IF values (i.e., within a truncated IF range) yields the best performance. They further propose a new data selection strategy to avoid the heavy computational cost of IF estimation, and demonstrate positive experimental results.

**Strengths:**

1. The idea is novel and surprising. Traditionally, it is believed that maximizing the margin (i.e., the loss difference) for single-step optimization is most beneficial for learning. However, this paper shows that truncated medium IF actually performs best.

2. The authors also carefully consider the high computational cost of IF computation in practice and propose an approximation strategy.

3. In the preliminary experiment, both DPO training and validation reward margins are computed on subsets from UltraFeedback, which share similar distributions, while in the main experiments, OOD situations are also tested, and the core design of this paper works on both situations, proving its value.

**Weaknesses:**

1. Although the analysis is thorough, the method itself is relatively simple, and the level of technical innovation is somewhat limited.

2. The proposed LossDiff-IRM takes the intersection of two selection criteria. However, model performance also depends on data scale. If the intersection yields a very small subset, it may pose a risk of data scarcity, which should be addressed.

3. Since pairwise preference data and DPO-style training are becoming less common, the strong coupling between the proposed data selection and model completions might limit its applicability to more recent paradigms such as RL-based methods. It would be helpful to include a discussion on how such approaches might extend to the prompt-level selection process.

**Questions:**

How are the thresholds set? I did not find sufficient details in the paper regarding the choice or tuning of these thresholds.

---

> ### Author Response · Authors · 2025-11-25
> **Response to reviewer CD9n (Part 1/3)**
>
> We sincerely thank the reviewer for the time and effort spent reviewing our paper and for acknowledging the novel and surprising idea, careful consideration of computational cost, and comprehensive experimental validation. Below, we address the raised concerns and questions in detail:
>
> > **W1:** Although the analysis is thorough, the method itself is relatively simple, and the level of technical innovation is somewhat limited.
>
> **A:** Thanks for your valuable comments. We would like to clarify that the core contributions of our work do not lie solely in proposing a new data-selection method. Instead, **the main innovation comes from using influence function (IF) as an analysis tool to analyze the preference data from viewpoint of generalization and offer convicing insights**. Through this analysis, we offer a novel observation: **preference pairs with medium-IF contribute most effectively to LLM alignment**. This finding offers a new understanding of how preference data influences preference generalization after alignment training.
>
> Building upon this finding, **we introduce a simple and intuitive data selection strategy (LossDiff-IRM), whose effectiveness is validated by extensive experiments across three LLM families and multiple model scales**. We think that the simplicity here is an advantage rather than a limitation: the ideas behind our analysis and method are easy to follow and can be extended into broader alignment approaches.
>
> > **W2:** The proposed LossDiff-IRM takes the intersection of two selection criteria. However, model performance also depends on data scale. If the intersection yields a very small subset, it may pose a risk of data scarcity, which should be addressed.
>
> **A:** Thanks for your insightful comments. We agree with the reviewer that data scale is an important factor in LLM training. We have not observed this sceneario in our experiments. In practice, **both LossDiff and IRM exhibit distributional shapes resembling unimodal, Gaussina-like distributions, where the majority of data lies in the middle region and only a small number of data falls in the extremes**. In our data selection strategy, both creteria concentrate data in their middle ranges, taking their intersection would not result in an extremely small subset. The overlap between their medium regions remains sufficiently large and high-quality for alignment training. Moreover, our experimental results further validate the effectiveness of the proposed LossDiff-IRM selection strategy: despite using fewer preference pairs, the performance is better than full-data training across multiple LLM families and benchmarks. To make this distributional property more intuitive, we will include visualizations of the LossDiff and IRM distributions in the revised manuscript.

---

> ### Author Response · Authors · 2025-11-25
> **Response to reviewer CD9n (Part 2/3)**
>
> > **W3:** Since pairwise preference data and DPO-style training are becoming less common, the strong coupling between the proposed data selection and model completions might limit its applicability to more recent paradigms such as RL-based methods. It would be helpful to include a discussion on how such approaches might extend to the prompt-level selection process.
>
> **A:** Thanks for your thoughtful comments. The reviewer's understanding is correct that our current analysis and method primarily focus on pairwise preference data and DPO-style training. **It is worth noting that our influence function (IF)-based analysis and data-selection strategy do not introduce extreme assumptions**. In addition to DPO and SLiC validated in our experiments, they can be naturally extended to a broader range of pairwise preference optimization methods such as CPO [1], SimPO [2], and RePO [3].
>
> We also agree with the reviewer that RL-based post-training paradigms, especially rule-based RL methods used in verifiable tasks such as math and code reasoning, have become increasingly prominent. However, for open-ended tasks such as preference alignment, pairwise preference data remain essential because reward modeling continues to rely heavily on pairwise comparisons to construct reward signals for online RL training. In this context, **the IF-based analysis and data-selection strategy can naturally extend to improving the quality of preference data used for reward modeling, thereby benefiting RL-based alignment pipelines as well**.
>
> Furthermore, since IF is a general and widely used analytical tool in machine learning, it has the potential to analyze other LLM training paradigms, such as pretraining [4] and supervised fine-tuning (SFT) [5]. We appreciate that the reviewer points out a promising future research direction to explore IF-based analysis for RL-based learning objectives such as PPO and GRPO.
>
> [1] Xu H, Sharaf A, Chen Y, et al. Contrastive Preference Optimization: Pushing the Boundaries of LLM Performance in Machine Translation[C]//Forty-first International Conference on Machine Learning.
> [2] Meng Y, Xia M, Chen D. Simpo: Simple preference optimization with a reference-free reward[J]. Advances in Neural Information Processing Systems, 2024, 37: 124198-124235.
> [3] Wu J, Huang K, Wang X, et al. RePO: Understanding Preference Learning Through ReLU-Based Optimization. NeurIPS, 2025.
> [4] Grosse R, Bae J, Anil C, et al. Studying large language model generalization with influence functions[J]. arXiv preprint arXiv:2308.03296, 2023.
> [5] Xia M, Malladi S, Gururangan S, et al. LESS: Selecting Influential Data for Targeted Instruction Tuning[C]//International Conference on Machine Learning. PMLR, 2024: 54104-54132.

---

> ### Author Response · Authors · 2025-11-25
> **Response to reviewer CD9n (Part 3/3)**
>
> > **Q1:** How are the thresholds set? I did not find sufficient details in the paper regarding the choice or tuning of these thresholds.
>
> A: Thanks for your valuable question. In our experiments, the percentile thresholds of $\xi_\text{small}, \xi_\text{large}, \tau_\text{small}, \tau_\text{larege}$ for LossDiff and IRM are tuned as hyperparameters. Specifically, we vary the low percentile thresholds $\xi_\text{small},\tau_\text{small}$ within {0, 10, 20, 30} and large percentile thresholds $\xi_\text{large},\tau_\text{large}$ within {100, 90, 80, 70}. To reduce the hyperparameter search space, we vary $\xi_\text{small}=(100-\xi_\text{large})$ and $\tau_\text{small}=(100-\tau_\text{large})$ within {0, 10, 20, 30}. The results of WinRate evaluated on UltraFeedback are reported as follows:
> |$ξ_\text{small} = (1 − ξ_\text{large})$|UltraFeedback||AlpacaEval||Vicuna-Bench||Arena-Hard||
> |-|-|-|-|-|-|-|-|-|
> |                           | Single        | WinRate   | Single     | WinRate   | Single       | WinRate   | Single     | WinRate   |
> |               |           |            | **Llama-3.1-8B**          |              |           |            |           |
> | 0                         | 5.77          | 77.61     | 5.87       | 78.41     | 6.04         | 73.75     | 4.68       | 81.39     |
> | 10                        | 6.34          | 82.42     | 6.49       | 84.36     | 6.47         | 79.69     | 5.32       | 87.29     |
> | 20                        | **6.47**      | **83.82** | **6.73**   | **84.50** | **6.76**     | **80.31** | **5.40**   | **88.38** |
> | 30                        | 6.24          | 82.26     | 6.38       | 81.74     | 6.54         | 77.68     | 4.64       | 81.27     |
> |      |               |           |    **Qwen3-8B-Base**         |           |              |           |            |           |
> | 0                         | 7.64          | 61.41     | 7.92       | 63.85     | 8.21         | 62.14     | 7.58       | 59.61     |
> | 10                        | 7.89          | 64.71     | **8.26**  | **68.10**   | **8.66** | **65.62**     | 7.72  | **64.06**    |
> | 20                        | **7.97**      | **65.19** | 8.23  | 67.96 | 8.31  | 63.36| **7.76**   | 63.42 |
> | 30                        | 7.70          | 63.78     | 8.08       | 64.52     | 8.09         | 61.25     | 7.48       | 61.32     |
>
> |  $\tau_\text{small} = (1 − \tau_\text{large})$ | UltraFeedback |           | AlpacaEval |           | Vicuna-Bench |           | Arena-Hard |           |
> |---------------------------|---------------|-----------|------------|-----------|--------------|-----------|------------|-----------|
> |                           | Single        | WinRate   | Single     | WinRate   | Single       | WinRate   | Single     | WinRate   |
> |               |           |            | **Llama-3.1-8B**          |              |           |            |           |
> | 0                         | 5.77          | 77.61     | **5.87**       | 78.41     | 6.04         | 73.75     | 4.68       | 81.39     |
> | 10                        | 5.72          | 76.92     | 5.80       | 77.25     | 5.97         | 76.88     | **4.86**   | **81.55** |
> | 20                        | **5.89**      | **78.52** | 5.84    | 77.68     | **6.39**     | **77.96** | 4.64       | 80.80     |
> | 30                        | 5.53          | 76.57     | 5.65| **78.64** | 5.96         | 76.88     | 4.50| 79.15 |
> |        |               |           |     **Qwen3-8B-Base**         |           |              |           |            |           |
> | 0                         | **7.64**| 61.41     | 7.92       | 63.85     | 8.21         | **62.14**     | 7.58       | 59.61     |
> | 10                        | 7.61          | **61.73** | 7.94       | 63.88     | **8.22**     | 59.38     | **7.62**       | 63.23     |
> | 20                        | 7.60          | 61.04     | 8.00       | 65.34 | 7.89         | 57.50     | 7.61 | **67.40** |
> | 30                        | **7.64**      | 60.63 | **8.01**   | **65.83**     | 8.07         | 59.38 | 7.61       | 65.81     |
>
> From the above results, it can be observed a rough trend: **performance first improves and then degrades as the thresholds become stricter, that is, as more data are filtered out**. This observation is expected, because initially, the threshold helps remove low-quality data; however, beyond a certain point, the filtering starts to exclude informative and high-quality data, leading to performance degradation. Finally, we set $\xi_\text{small}$ and $\tau_\text{small}$ as 10, $\xi_\text{large}$ and $\tau_\text{large}$ as 90 for all models. We will include these results in the revised manuscript.

---

> ### Author Response · Authors · 2025-11-28
> **Looking forward to your reply**
>
> Dear Reviewer CD9n,
>
> We sincerely thank you for your efforts in reviewing our work and for your great support!
>
> Although you may not update your evaluation due to the unexpected accident, we would be very glad to engage in further discussion with you if you have further concerns or questions. You constructive feedback truly helps us a lot in improving the quality of the work
>
> Thank you once again for your valuable time and efforts.
>
> Authors of # 13211

---

### Official Review · Reviewer_8ymy · 2025-11-01

**Soundness:** 2
**Presentation:** 3
**Contribution:** 2
**Rating:** 4
**Confidence:** 4

**Summary:**

The paper reframes preference-data quality as model-dependent and introduces a Truncated Influence Function (TIF) lens showing that medium-IF pairs—not very small or very large ones—drive the most stable alignment gains. To avoid the high cost of exact IF on LLMs, the authors propose two forward-pass proxies—Loss Difference (LossDiff) and Implicit Reward Margin (IRM)—and a combined LossDiff–IRM selector that closely tracks TIF.

**Strengths:**

1. **Principled, model-aware data valuation.** The paper grounds selection in a TIF-motivated view of *model-dependent* data value and shows that combining **LossDiff** and **IRM** outperforms either proxy alone; moreover, pairs discarded by the selector are empirically low-value for alignment, confirming the criterion’s discriminative power.

2. **Strong empirical generality.** Across models, objectives (DPO/SLiC), and benchmarks, the method achieves higher win rates using reduced data (e.g., 64% subset with top performance).

**Weaknesses:**

1. **Incomplete positioning vs. recent work.** The paper discusses Filtered DPO, but does not engage with several highly relevant baselines:

   * [1] margin-based preference selection for alignment quality (ICML 2025),
   * [2] RS-DPO (rejection sampling + DPO for cleaner preference data),
   * [3] active preference learning for LLMs (querying informative pairs instead of passively filtering).
     These works target the same core problem — selecting / curating high-value preference pairs — and should be compared both conceptually and empirically.

2. **Practicality.** The proposed selectors still require rescoring large volumes of pairs with both the current model and an auxiliary model. The paper does not report the real cost (GPU hours, throughput) or show that this is cheaper/more scalable than RS-DPO-style sampling or active preference acquisition.

3. **Robustness.** The “medium-IF is best” claim is convincing on the reported setups but is not stress-tested across broader domains, model sizes, or stages of alignment; it is unclear how stable this curriculum is outside the presented benchmarks.

[1] Larger or Smaller Reward Margins to Select Preferences for Alignment? ICML 2025.
[2] Rs-dpo: A hybrid rejection sampling and direct preference optimization method for alignment of large language models. NAACL 2024.
[3] Active preference learning for large language models. ICML 2024.

**Questions:**

See weaknesses.

---

> ### Author Response · Authors · 2025-11-25
> **Response to reviewer 8ymy (Part 1/3)**
>
> We sincerely thank the reviewer for the time and effort spent reviewing our paper and for acknowledging the principled data valuation and strong empirical generality. Below, we address the raised concerns and questions in detail:
>
> > **W1:** Incomplete positioning vs. recent work. The paper discusses Filtered DPO, but does not engage with several highly relevant baselines:
> [1] margin-based preference selection for alignment quality (ICML 2025),
> [2] RS-DPO (rejection sampling + DPO for cleaner preference data),
> [3] active preference learning for LLMs (querying informative pairs instead of passively filtering). These works target the same core problem — selecting / curating high-value preference pairs — and should be compared both conceptually and empirically.
>
> **A:** Thanks for your valuable comments that providing these related studies. We carefully review these papers and appreciate the reviewer that these studies are very relevant to our work. **We have cited them in our revised manuscript**. Specifically, Huang et al. [1] propose the alignment potential $M_\text{AP}$ that integrates both explicit and implicit reward margins to quantify preference data. Khaki et al. [2] propose the RS-DPO that combines rejection sampling (RS) with DPO to generate preference data for DPO, where an external reward model is required to score preference pairs during RS. Muldrew et al. [3] propose an active learning framework for DPO that tends to select informative prompt-completion pairs for labelling based on predictive entropy and implicit preference certainty. Overall, [1] focuses on reward margin-based data selection, whereas [2] and [3] mainly focus on preference pair generation.
>
> Beyond these methods, our work makes a distinct and complementary contribution. **Rather than only proposing another data selection approach, we build a comprehensive analytical pipeline that investigates preference data from the viewpoint of the influence function (IF).** We employ IF as a principled analytical tool, aiming to understand which types of preference pairs truly contribute to preference generalization during alignment training. This analysis offers a novel observation: **preference pairs with medium-IF contribute more effectively for LLM alignment**. Our proposed data selection strategy is directly grounded in these IF-driven insights, and these insights may inspire other analogous selection strategies. Moreover, our analysis and selection approach **do not rely on external reward models** used in [1,2] **nor on additional annotation** required by [3].
>
> Following the reviewer's suggestion, **we further include several baselines for more comprehensive comparisons, including RS-DPO [2], $M_\text{AP}$[1], and CurriDPO [4]**. CurriDPO orders preference pairs to organize curriculum learning based on creteria, such as GPT4 score and reward model score. The results are:
> |Llama-3.1-8B|UltraFeedback Single|WinRate|AlpacaEval Single|WinRate|Vicuna-Bench Single|WinRate|Arena-Hard Single|WinRate|
> |-|-|-|-|-|-|-|-|-|
> |CurriDPO-GPT4|5.47|74.23|5.53|75.01|5.84|74.06|4.55|79.77|
> |CurriDPO-Reward Model|5.51|74.62|5.54|74.29|5.59|74.06|4.62|79.49|
> |$M_\text{AP}$|6.04|79.99|6.21|80.88|6.34|74.69|5.08|85.30|
> |RS-DPO|5.70|75.98|6.39|84.84|7.04|82.81|4.69|82.05|
> |**LossDiff-IRM (Ours)**|**6.54**|**83.97**|**6.84**|**87.08**|**7.06**|**86.88**|**5.59**|**88.40**|
>
> |Qwen3-8B-Base|UltraFeedback Single|WinRate|AlpacaEval Single|WinRate|Vicuna-Bench Single|WinRate|Arena-Hard Single|WinRate|
> |-|-|-|-|-|-|-|-|-|
> |CurriDPO-GPT4|7.61|61.04|7.74|62.35|8.16|54.69|7.52|62.51|
> |CurriDPO-Reward Model|7.60|59.62|7.84|61.96|8.20|61.58|7.55|63.97|
> |$M_\text{AP}$|7.95|**67.84**|8.31|71.11|8.62|66.87|7.72|66.55|
> |RS-DPO|7.88|64.87|**8.36**|**74.07**|**8.93**|**71.90**|7.48|61.32|
> |**LossDiff-IRM (Ours)**|**8.05**|67.32|**8.36**|71.52|8.72|67.19|**7.83**|**68.63**|
>
> We observe that **LossDiff-IRM outperforms these baselines on most cases**. Compared to CurrDPO relied on external signals (GPT4 or reward model scores), $M_\text{AP}$ and RS-DPO, which adopt implicit reward margins or generate model-specific preference pairs using the SFT model and thus partially incorporate the model’s perspective, emerge as strong competitors. This observation aligns with the underlying philosophy of our analysis: valuable preference data are model-dependent, rather than relying on external heuristics. **These results further validate the effectiveness of our IF-inspired analysis and LossDiff-IRM data selection method**. We will include these results and corresponding discussion in the revised manuscript.
>
> [1] Larger or Smaller Reward Margins to Select Preferences for LLM Alignment? ICML 2025.
> [2] RS-DPO: A Hybrid Rejection Sampling and Direct Preference Optimization Method for Alignment of Large Language Models. NAACL 2024.
> [3] Active Preference Learning for Large Language Models. ICML 2024.
> [4] Enhancing alignment using curriculum learning & ranked preferences. EMNLP 2024.

---

> ### Author Response · Authors · 2025-11-25
> **Response to reviewer 8ymy (Part 2/3)**
>
> > **W2:** Practicality. The proposed selectors still require rescoring large volumes of pairs with both the current model and an auxiliary model. The paper does not report the real cost (GPU hours, throughput) or show that this is cheaper/more scalable than RS-DPO-style sampling or active preference acquisition.
>
> **A:** Thanks for your insightful comments. We would like to clarify that RS-DPO [2] primarily focuses on **generating new high-quality preference pairs** through rejection sampling using the SFT model, whereas our work focuses on **analyzing existing annotated preference pairs** using an influence function (IF)-based analytical pipeline to identify valuable preference pairs. **These two paradigms differ fundamentally in their purpose and cost structures**.
>
> Moreover, RS-DPO requires **an external reward model** to evaluate the responses generated by the SFT model. Given N prompts and K sampled responses per prompt, **RS-DPO needs to score N×K responses with the reward model** in order to construct contrastive preference pairs. For LossDiff-IRM, the reviewer's understanding is correct that **LossDiff-IRM requires scoring all preference pairs** with both the current model and an auxiliary model. For the same scale of N preference pairs, LossDiff-IRM requires N forward passes with the current model and N forward passes with the auxiliary model, **resulting in a total of 2N forward passes**, where the two models share the same architecture and differ only in parameters.
>
> To facilitate a direct comparison, we report below the **data processing time cost of RS-DPO** (including data generation and rejection sampling) using the OpenAssistant-1.4B reward model with K=8, as well as the **data-selection time of LossDiff-IRM**. All experiments were conducted on a single H100-80GB GPU:
>
> | Trained Model   | Number of Prompts | RS-DPO Time   | RS-DPO Throughput (prompts/sec) | LossDiff-IRM Time | LossDiff-IRM Throughput |
> |------------------|--------------------|----------------|----------------------------------|---------------------|---------------------------|
> | Llama-3.1-8B     | 48,908             | 4 h 34 min     | 2.97                             | 2 h 14 min          | 6.08                      |
> | Qwen3-8B-Base    | 48,908             | 4 h 50 min     | 2.81                             | 2 h 43 min          | 5.03                      |
>
> The dominant cost of RS-DPO comes from using the reward model to evaluate all N x K responses generated by the SFT model. Even using a relatively small 1.4B reward model, the time cost is actually higher than LossDiff-IRM. Therefore, **the time cost of LossDiff-IRM is at least on par with RS-DPO, and even often smaller**, especially when considering that RS-DPO scales linearly with K or uses a larger reward model.

---

> ### Author Response · Authors · 2025-11-25
> **Response to reviewer 8ymy (Part 3/3)**
>
> > **W3:** Robustness. The “medium-IF is best” claim is convincing on the reported setups but is not stress-tested across broader domains, model sizes, or stages of alignment; it is unclear how stable this curriculum is outside the presented benchmarks.
>
> **A:** Thanks for your thoughtful comments. We agree that **incorporating more benchmarks and model sizes is important for improving the robustness of our conclusions. Our experimental design has followed this principle:** we conduct experiments across three LLM families (Qwen3, Llama3 and Pythia) with model scales spanning 410M, 1.4B, 2.8B and 8B. In addition, our analysis pipeline is carefully structured. We begin with two smaller models to clearly offer the key observation that preference pairs with medium-valued influence function (IF) contribute most effectively to alignment training. Building on this motivation, we design our data-selection strategy and then validate it on multi-scale, multi-family LLMs. The experimental results reasonably support our analysis and validate its effectiveness.
>
> Following the reviewer's suggestion, **we additionally supplement the evaluation on the Arena-Hard benchmark across all models trained in our manuscript to enhance the robustness of our experiments**. The results are summarized as follows:
>
> |Method|Llama-3.1-8B (DPO)||Qwen3-8B-Base (DPO)||Pythia-2.8B (DPO)||Pythia-1.4B (DPO)||Pythia-410M (DPO)||
> |-|-|-|-|-|-|-|-|-|-|-|
> ||**Single**|**WinRate**|**Single**|**WinRate**|**Single**|**WinRate**|**Single**|**WinRate**|**Single**|**WinRate**|
> |**SFT**|2.63|-|6.64|-|2.74|-|2.37|-|1.91|-|
> |**Full Data**|4.68|81.39|7.58|59.61|2.97|60.71|2.65|60.24|2.06|59.47|
> |**Random**|4.64|81.27|7.57|62.07|3.00|63.25|2.60|61.64|2.06|57.08|
> |**GPT4**|4.96|84.30|7.62|61.53|3.08|64.15|2.83|64.20|2.15|61.44|
> |**Reward Model**|5.13|86.19|7.61|64.78|3.03|63.03|2.71|64.76|2.16|60.59|
> |**LossDiff-IRM**|**5.59**|**88.40**|**7.83**|**68.63**|**3.26**|**71.64**|**2.96**|**71.72**|**2.38**|**69.63**|
>
> |Method|Llama-3.1-8B (SLiC)||Qwen3-8B-Base (SLiC)||Pythia-2.8B (SLiC)||Pythia-1.4B (SLiC)||Pythia-410M (SLiC)||
> |-|-|-|-|-|-|-|-|-|-|-|
> ||**Single**|**WinRate**|**Single**|**WinRate**|**Single**|**WinRate**|**Single**|**WinRate**|**Single**|**WinRate**|
> |**SFT**|2.63|-|6.64|-|2.74|-|2.37|-|1.91|-|
> |**Full Data**|3.98|73.75|7.21|59.61|2.93|59.04|2.65|58.95|2.03|57.49|
> |**Random**|3.95|70.35|7.25|59.18|2.98|57.58|2.66|60.26|2.14|59.77|
> |**GPT4**|4.28|75.27|7.33|59.47|2.89|58.38|2.68|59.80|2.15|60.07|
> |**Reward Model**|4.50|78.32|7.38| **62.27** |2.91|59.18|2.67|61.95|2.15|60.11|
> |**LossDiff-IRM**|**4.65**|**83.12**|**7.61**|62.20|**3.19**|**64.83**|**2.85**|**67.76**|**2.21**|**62.40**|
>
> From the above results, we observe similar performance trends as evaluated on the other three benchmarks (UltraFeedback, AlpacaEval and Vicuna-Bench): **our LossDiff-IRM outperforms full-data training and data selection by GPT4 or reward model on most cases**. The additional evaluation on Arena-Hard further indicates the effectiveness of LossDiff-IRM in identifying valuable preference data for alignment training. **These results will be incorporated into the revised manuscript**.

---

> ### Comment · Reviewer_8ymy · 2025-11-27
>
> Thanks for the detailed rebuttal and the additional experiments. After reading them, I’m updating my score to 6.
>
> The new comparisons with $M_{AP}$ and RS-DPO make the positioning of LossDiff-IRM much clearer, and the compute breakdown helped address my concerns about practicality relative to generation-based approaches. I’d strongly encourage keeping these parts in the final version—they do a lot of work in clarifying the contribution.
>
> One thing I’m still curious about: do you expect the "medium-IF" band to move as training progresses? In other words, might a pair that looks "medium/valuable" early on become "easy/low-IF" after a few epochs? It would be interesting to see whether a dynamic or iterative selection strategy could further improve over a single static filtering pass.

---

> > ### Author Response · Authors · 2025-11-27
> > **Thank you**
> >
> > Dear Reviewer 8ymy,
> >
> > Thank you very much for your continued support and insightful comments, which means a lot to us. **We sincerely appreciate your reconsideration for raising the score, and we are grateful for your constructive feedback throughout the review process**.
> >
> > Regarding your insightful question about whether the “medium-IF” band may shift as training progresses, yes, this may occur, because the model’s gradients evolve over training. In response to this point, we compute the pairwise Jaccard Correlation Coefficients between the preference data selected by IF using checkpoints from epochs {0, 1, 2, 3, 4, 5} of Qwen3-0.6B-Base. The results are reported as follows:
> > |Jaccard Coefficient|0|1|2|3|4|5|
> > |-|-|-|-|-|-|-|
> > |**0**|1.00|0.90|0.86|0.84|0.83|0.83|
> > |**1**||1.00|0.94|0.91|0.89|0.89|
> > |**2**|||1.00|0.96|0.94|0.94|
> > |**3**||||1.00|0.98|0.97|
> > |**4**|||||1.00|0.99|
> > |**5**||||||1.00|
> >
> > These results show that although the correlation gradually decreases as the epoch gap increases, all coefficients remain above 0.8. This indicates that **the medium-IF band does shift during training, but only to a limited extent**. Together with Figure 5 in our manuscript, we also observe that the shift across different training stages of the same model is noticeably smaller than the shift across different models, suggesting the model-specific nature of IF to assess preference data. We note that this analysis is a preliminary exploration, and your comments raises a very interesting point. Integrating IF-driven data analysis and selection into training dynamics would be a promising future research direction.
> >
> > Thank you once again for your valuable time and efforts!
> >
> > Best regards,
> >
> > Authors of # 13211

---

### Official Review · Reviewer_kSkJ · 2025-11-11

**Soundness:** 3
**Presentation:** 2
**Contribution:** 2
**Rating:** 6
**Confidence:** 2

**Summary:**

This paper studies the problem of identifying valuable preference data for aligning large language models (LLMs) with human preferences. The authors argue that data quality should be viewed as a model-dependent property, not intrinsic to the data itself, and further introduce the Truncated Influence Function (TIF) to evaluate the per-sample effect of preference pairs on validation performance, showing that “medium-IF” data tend to yield the best alignment results.

To make this computation feasible at scale, they propose two lightweight proxies—Loss Difference (LossDiff) and Implicit Reward Margin (IRM)—and a combined rule LossDiff–IRM for preference data selection. Experiments on various LLM models as well as alignment methods show that the proposed selection could achieve better alignment with about half of the dataset, improving winrate over full-data baselines.

**Strengths:**

1. This paper is well-motivated. The idea that data valuation is model-dependent is insightful and challenges a long-standing assumption in RLHF and preference optimization pipelines.


2. The proposed Truncated Influence Function (TIF) seems new in the literature and provides a principled mechanism to quantify individual data influence.

3. Results span several LLM families, datasets, and alignment methods, demonstrating generality and robustness.

**Weaknesses:**

1. The “medium-IF” hypothesis is interesting, but primarily from the empirical observations. It would be better if the authors could elaborate more in a theoretical or formal way.

2.  It seems a commonly used benchmark, arena-hard, is not included. Can authors elaborate more on it?

3. (Mirror Issue) The presentation could also be improved. For example, the captions of Figure 1 and Figure 3 refer to the materials in the appendix. From my personal perspective, it might be better to put the related paragraphs back into the main paper.

**Questions:**

Please see the weakness section.

In summary, this paper provides a well-motivated perspective on preference data valuation, supported with solid empirical evidence. I currently tend to recommend acceptance. However, I'm willing to re-evaluate this work according to the further dicusssions.

---

> ### Author Response · Authors · 2025-11-25
> **Response to reviewer kSkj (Part 1/2)**
>
> We sincerely thank the reviewer for the time and effort spent reviewing our paper and for acknowledging the well-motivated and insightful idea, novelty of our TIF, and comprehensive experiments. Below, we address the raised concerns and questions in detail:
>
> > **W1:** The "medium-IF" hypothesis is interesting, but primarily from the empirical observations. It would be better if the authors could elaborate more in a theoretical or formal way.
>
> **A:** Thanks for your valuable comments. Following the reviewer's suggestion, we demonstrate that the reliability of selecting high‑influence function (IF) data decreases as the quality of the validation data decreases. We provide the derivation pipeline as follows:
>
> Suppose the validation dataset is corrupted w.p. $\eta$. **Then, the expected validation gradient $v$ consists of the clean ($c$) and noisy ($n$) parts, following  $g_v=(1-\eta)g_c+\eta g_n$**. For a training sample $i$ with the gradient $g_i$, we have its influence weight as $\omega_i=-g_v^\top g_i=-(1-\eta)g_c^\top g_i-\eta g^\top_n g_i$, which can be further written as $\omega_i=\omega^{\*}_i+\delta_i$, with $\omega^{\*}_i$ the clean weight and $\delta_i=-\eta(g_n-g_c)^\top g_i$ the bias term. Assuming that $g_n-g_c$ is independent on $g_i$, with zero mean and isotropic variance $\sigma_n^2 I$, then we have $\text{Var}(\delta_i)=\eta^2\sigma^2_n\vert\vert g_i\vert\vert^2$ and $\text{Var}(\omega_i)=\sigma_w^2+\tau^2$, with $\sigma_w^2=\text{Var}(\omega^{\*}_i)$ and $\tau^2=\text{Var}(\delta_i)$. Accordingly, we have the expected correlation between $\omega^{\*}$ and $\omega$ as $\rho={\text{Cov}(\omega^{\*}_i,\omega_i)}/{\sqrt{\text{Var}(\omega^\{*}_i)\text{Var}(\omega_i)}}=\sqrt{{\sigma^2_w}/{\sigma^2_w+\tau^2}}$.
>
> Now, we test the reliability of the top-$k$ sampling strategy under corruption. Let $\mathcal T(k)$ denote the set of indices whose $\omega$ values fall above the top-$k$ quantile, and let $z_k$ be the corresponding $z$-score threshold satisfying $p(\omega_i>z_k)=k$. **We are interested in the probability that an sample belongs to the true top-$k$ set $\mathcal T_{\text{true}}$ given that it is observed to be in the niosy top-$k$ set $\mathcal T_{\text{obs}}$**, which is given by $p(i\in\mathcal T_{\text{true}}(k)|i\in\mathcal T_{\text{obs}}(k))=p(\omega_i^{\*}>z_k|\omega_i>z_k)$. Based on the large deviation theories, we can estimate $p(\omega_i^{\*}>z_k,\omega_i>z_k)$ as $(1-\Phi(z_k))\Phi(-\frac{z_k(1-\rho)}{\sqrt{1-\rho^2}})=k\Phi(-z_k\sqrt{\frac{1-\rho}{1+\rho}})$, and then we have $p(i\in\mathcal T_{\text{true}}(k)|i\in\mathcal T_{\text{obs}}(k))=\frac{p(\omega_i^*>z_k,\omega_i>z_k)}{p(\omega_i>z_k)}=\Phi(-z_k\sqrt{\frac{1-\rho}{1+\rho}})$.
>
> As observed, **when the corruption rate $\eta$ increases, the correlation $\rho$ between the true and observed scores decreases, leading to a lower probability $p(i\in\mathcal T_{\text{true}}(k)|i\in\mathcal T_{\text{obs}}(k))$**. Hence, the top‑$k$ sampling strategy becomes less reliable under high noise levels. Similar derivations hold for bottom‑$k$ sampling, making middle‑$k$ sampling  tend to be more reliable. We will add the elaboration into the revised manuscript and hope this can present our insight formally.

---

> ### Author Response · Authors · 2025-11-25
> **Response to reviewer kSkj (Part 2/2)**
>
> > **W2:** It seems a commonly used benchmark, arena-hard, is not included. Can authors elaborate more on it?
>
> **A:** Thanks for your insightful comments. We agree with the reviewer that Arena-Hard is also a commonly used benchmark in LLM alignment, and including it is helpful to solidite our experiments. Following the reviewer's suggestion, **we supplement the evaluation on Arena-Hard benchmark across all models trained in our manuscript**. The results are reported as follows:
> |Method|Llama-3.1-8B (DPO)||Qwen3-8B-Base (DPO)||Pythia-2.8B (DPO)||Pythia-1.4B (DPO)||Pythia-410M (DPO)||
> |-|-|-|-|-|-|-|-|-|-|-|
> ||**Single**|**WinRate**|**Single**|**WinRate**|**Single**|**WinRate**|**Single**|**WinRate**|**Single**|**WinRate**|
> |**SFT**|2.63|-|6.64|-|2.74|-|2.37|-|1.91|-|
> |**Full Data**|4.68|81.39|7.58|59.61|2.97|60.71|2.65|60.24|2.06|59.47|
> |**Random**|4.64|81.27|7.57|62.07|3.00|63.25|2.60|61.64|2.06|57.08|
> |**GPT4**|4.96|84.30|7.62|61.53|3.08|64.15|2.83|64.20|2.15|61.44|
> |**Reward Model**|5.13|86.19|7.61|64.78|3.03|63.03|2.71|64.76|2.16|60.59|
> |**LossDiff-IRM**|**5.59**|**88.40**|**7.83**|**68.63**|**3.26**|**71.64**|**2.96**|**71.72**|**2.38**|**69.63**|
>
> |Method|Llama-3.1-8B (SLiC)||Qwen3-8B-Base (SLiC)||Pythia-2.8B (SLiC)||Pythia-1.4B (SLiC)||Pythia-410M (SLiC)||
> |-|-|-|-|-|-|-|-|-|-|-|
> ||**Single**|**WinRate**|**Single**|**WinRate**|**Single**|**WinRate**|**Single**|**WinRate**|**Single**|**WinRate**|
> |**SFT**|2.63|-|6.64|-|2.74|-|2.37|-|1.91|-|
> |**Full Data**|3.98|73.75|7.21|59.61|2.93|59.04|2.65|58.95|2.03|57.49|
> |**Random**|3.95|70.35|7.25|59.18|2.98|57.58|2.66|60.26|2.14|59.77|
> |**GPT4**|4.28|75.27|7.33|59.47|2.89|58.38|2.68|59.80|2.15|60.07|
> |**Reward Model**|4.50|78.32|7.38| **62.27** |2.91|59.18|2.67|61.95|2.15|60.11|
> |**LossDiff-IRM**|**4.65**|**83.12**|**7.61**|62.20|**3.19**|**64.83**|**2.85**|**67.76**|**2.21**|**62.40**|
>
> From the above results, we observe that the performance trends on Arena-Hard across different models and alignment approaches are consistent with those on the other three benchmarks (UltraFeedback, AlpacaEval, and Vicuna-Bench), where **LossDiff-IRM outperforms full-data training and data data selection strategies based on GPT-4 score or external reward model on most cases**. The evaluation on Arena-Hard further demonstrates the effectiveness of LossDiff-IRM in identifying valuable preference data for alignment training. We will update these results in the revised manuscript.
>
> > **W3:** (Mirror Issue) The presentation could also be improved. For example, the captions of Figure 1 and Figure 3 refer to the materials in the appendix. From my personal perspective, it might be better to put the related paragraphs back into the main paper.
>
> **A:** Thanks for your helpful suggestion regarding the presentation. We would like to express our sincere thanks to the reviewer's feedback on improving the writing quality. We will carefully check and proofread the full manuscript. Following the reviewer's suggestion, **we will strengthen the cross-referencing between the main paper and the appendix to ensure smoother navigation between related paragraphs**. We fully agree with the reviewer that these improvements will enhance the clarity and readability of the manuscript.

---

> ### Author Response · Authors · 2025-11-28
> **Looking forward to your reply**
>
> Dear Reviewer kSkJ,
>
> We sincerely thank you for your efforts in reviewing our work and for your great support!
>
> Although you may not update your evaluation due to the unexpected accident, we would be very glad to engage in further discussion with you if you have further concerns or questions. You insightful comments truly help us a lot in improving the quality of the work
>
> Thank you once again for your valuable time and efforts.
>
> Authors of # 13211

---

### Author Response · Authors · 2025-11-27
**General response**

Dear All Reviewers and AC,

We would like to express our deep gratitude to all reviewers for their insightful suggestions and constructive comments, which are substantially helpful for us. We are also grateful for the multiple positive remarks highlighted across the reviews. In particular, we are pleased to see the following positive recognitions:
- **Well-motivated and clear idea:** kSkJ, 8ymy, CD9n, sQAd
- **Novelty of our influence function (IF)-driven analysis and proposed method:** kSkJ, 8ymy, CD9n, sQAd
- **Solid experiments across multiple LLM families and benchmarks:** kSkJ, 8ymy, sQAd
- **Strong empirical generality and performance:** kSkJ, 8ymy, CD9n, sQAd
- **Practicality of addressing the high cost of exact IF computation:** kSkJ, 8ymy, CD9n
- **Out-of-distribution (OOD) generalization:** CD9n
- **Well-written quality of our manuscript:** sQAd
- **Comprehensive ablation studies:** sQAd

More importantly, inspired by the reviewers' valuable comments and suggestions, our manuscript has been continually improved regarding the unclear parts or experiments about some specific points. We carefully followed the reviewers' suggestions to include additional experiments in our revised manuscript. For your reference, we supplement the main additions in the following:
- **Theoretical analysis of our IF-driven motivation:** supplemented in Appendix B.3.
- **Additional evaluation on benchmark Arena-Hard:** supplemented in Table 3.
- **Additional comparisons with several baselines of CurriDPO, MAP and RS-DPO:** supplemented in Table 4.
- **Additional analysis of percentile thresholds:** supplemented in Figure 6 and Figure 14.
- **Additional ablation study:** supplemented in Figure 3 and Figure 12.
- **Time cost analysis of RS-DPO and LossDiff-IRM:** supplemented in Table 14 and Appendix E.5.
- **Visualization of distribution of loss difference (LossDiff) and implicit reward margin (IRM):** supplemented in Figure 15 and Appendix E.7.

Thanks once again for all reviewers' valuable time and efforts. Your insigtful comments, no matter about the pros or the cons, all have been instrumental in improving the quality of our manuscript and have inspired us to continue advancing this research.

Best regards,

Authors of # 13211

---

### Author Response · Authors · 2025-12-02
**Summary and Clarification of the Rebuttal**

Dear Area Chairs,

We would like to express our deep gratitude for your valuable time and effort in handling our submission. We are writing to provide a comprehensive summary of our rebuttal process regarding **Submission 13211, which initially received scores of 6,6,6,4 (avg. 5.5) and finally achieved 6,6,6,6 (avg. 6.0) after valid discussion and before the information leakage incident**. We understand that the ACs face an extraordinary workload this year. To help save your valuable time, we would like to **clarify the timeline and recap our contributions, along with the substantial efforts we devoted during the rebuttal process for this work**.

**Clarification on Score Changes and Response Timeline:**
- According to the official report, this API-leakage event happened at 10:09 Nov 27 EST.
- Our submission initially receives **6,6,6,4 (avg. 5.5)**. After rebuttal, before the API-leakage event, the scores become **6,6,6,6 (avg. 6.0)**. This indicates that **all reviewers had reached a positive consensus** based on our rebuttal before the leakage incident occurred.
- **Reviewer 8ymy (4$\rightarrow$6) feedbacks positive comments** that we have adequately addressed all concerns, and **raises score from 6 to 8** at 03:48 Nov 27 EST, **before the leakage incident**.
- **We actively and continuously engaged in discussion with reviewer sQAd**, providing additional experiments and detailed clarifications at 15:52 on Nov 26 (EST) and 4:48 on Nov 28 (EST). Due to the leakage incident, the reviewer was unable to continue responding. From our discussion with the reviewer, **we observed that the reviewer's concerns were being gradually addressed** through our timely responses, additional experiments and detailed clarifications. Moreover, **the reviewer's initial score is positive, with a rating of 6**.
- Although we provided detailed point-by-point responses and left sufficient time for further discussions, two reviewers (kSkJ and CD9n) did not respond to us.

**Recapping the main contributions of our work**, we build a comprehensive analytical pipeline that investigates preference data from the viewpoint of the generalization. We employ influence function (IF) as a principled analytical tool, aiming to understand which types of preference pairs truly contribute to preference generalization during alignment training. This analysis offers a novel observation: preference pairs with medium-IF contribute more effectively for LLM alignment. Building on this finding and considering the heavy cost of exact IF computation, we propose the LossDiff-IRM data selection strategy, which approximates exact IF to identify high-quality preference pairs and further improve alignment performance. Empirically, we conduct extensive experiments across five models, four benchmarks, and two alignment approaches, demonstrating that our method achieves better performance while using less data.

**We are encouraged to receive reviewers' recognition for our core contribution:**
- **Well-motivated and clear idea:** All reviewers (kSkJ, 8ymy, CD9n, sQAd) recognize that our analysis and method are well-motivated.
- **Strong novelty:** All reviewers (kSkJ, 8ymy, CD9n, sQAd) recognize that our analysis provides a new insight to understand preference data.
- **Solid experiments:** Three reviewers (kSkJ, 8ymy, sQAd) acknowledge that our experiments are solid and comprehensive, spanning five models, four benchmarks, and two alignment approaches.
- **Strong performance:** All reviewers (kSkJ, 8ymy, CD9n, sQAd) notice that our method achieves strong empirical generality and performance over existing data selection strategy.

We **carefully provided point-by-point responses** to all reviewer comments, supplemented with **solid experiments and detailed clarifications**, which are outlined in [general response](https://openreview.net/forum?id=FUp0KeEEBs&noteId=knPbAuTO1W). Additionally, **we timely and actively engage in discussion with reviewers**. For example, reviewer sQAd replied at 8:05, 25 Nov EST, for additional experiments, and we imediately trained and evaluated the new experiments, and replied at 15:52, 26 Nov EST.

**Regarding reviewer 8ymy's current reverted score of 4 (a result of the incident)**, we had already addressed the reviewer's all concerns and questions before the leakage incident occurred. **The reviewer raised the score from 4 to 6. Therefore, we would like to emphasize that all reviewers had reached a positive consensus (6, 6, 6, 6) prior to the incident**. Unfortunately, the incident disrupted the normal rebuttal process and caused our substantial efforts to be undermined.

We deeply value all reviewers' feedback and sincerely appreciate the efforts of the area chairs. **We respectfully hope that both our academic contributions and the substantial efforts we have devoted to this work will be taken into proper consideration when the final decision is made.**

Best regards,

Authors of #13211

---

### Meta-Review · Area_Chair_r359 · 2026-01-07

**Summary:**

The main concerns influencing the decision:
- positioning/novelty: difference from prior preference-data selection work (especially margin/IRM-based and filtering methods). (8ymy)
- practicality/complexity: cost reporting for rescoring large corpora and comparison to sampling/active acquisition alternatives. (8ymy)
- robustness/generalization: the claim of the “medium-IF is best” hypothesis across domains/training stages.  (8ymy, sQAd, CD9n)
- presentation/detail issues: such as threshold setting, figure/caption dependence on appendix; missing/unclear benchmark coverage like Arena-Hard. (kSkJ)

**Reviewer Concerns:**

Addressed by the rebuttal:
- positioning vs. key baselines: the rebuttal added comparisons (MAP, RS-DPO), which clarified novelty/positioning. (8ymy)
- practicality: the rebuttal added the compute cost. (8ymy)
- baseline coverage/experimental details: additional experiments/clarifications/baselines were added. (8ymy, sQAd, CD9n)
- broader robustness/generalization: added experiments across more domains/model sizes/stages. (8ymy)
- presentation issues: solved. (kSkJ)

Still outstanding / should be noted as limitations or required fixes in camera-ready:
- applicability beyond DPO-style pairwise setups: discussion/extension to RL-based paradigms or prompt-level selection. (CD9n)

**Reviewer Scores:**

- kSkJ (original score 6): likely unchanged overall
- 8ymy (original score 4): already increased to 6 after rebuttal
- CD9n (original score 6): likely stable
- sQAd (original score 6): likely small upward improvements.

---

### Decision · Program_Chairs · 2026-01-26

Accept (Poster)